# Bisphenol A (BPA) and Cardiovascular or Cardiometabolic Diseases

**Jeong-Hun Kang [1],* , Daisuke Asai [2] and Riki Toita [3,4]**

[1] National Cerebral and Cardiovascular Center Research Institute, 6-1 Shinmachi, Kishibe, Osaka 564-8565, Japan

[2] Laboratory of Microbiology, Showa Pharmaceutical University, 3-3165 Higashi-Tamagawagakuen, Tokyo 194-8543, Japan; asai@ac.shoyaku.ac.jp

[3] Biomedical Research Institute, National Institute of Advanced Industrial Science and Technology (AIST), 1-8-31 Midorigaoka, Osaka 563-8577, Japan; toita-r@aist.go.jp

[4] AIST-Osaka University Advanced Photonics and Biosensing Open Innovation Laboratory, National Institute of Advanced Industrial Science and Technology (AIST), 2-1 Yamadaoka, Osaka 565-0871, Japan

\* Correspondence: jrjhkang@ncvc.go.jp

**Abstract:** Bisphenol A (BPA; 4,4′-isopropylidenediphenol) is a well-known endocrine disruptor. Most human exposure to BPA occurs through the consumption of BPA-contaminated foods. Cardiovascular or cardiometabolic diseases such as diabetes, obesity, hypertension, acute kidney disease, chronic kidney disease, and heart failure are the leading causes of death worldwide. Positive associations have been reported between blood or urinary BPA levels and cardiovascular or cardiometabolic diseases. BPA also induces disorders or dysfunctions in the tissues associated with these diseases through various cell signaling pathways. This review highlights the literature elucidating the relationship between BPA and various cardiovascular or cardiometabolic diseases and the potential mechanisms underlying BPA-mediated disorders or dysfunctions in tissues such as blood vessels, skeletal muscle, adipose tissue, liver, pancreas, kidney, and heart that are associated with these diseases.

**Keywords:** cardiovascular disease; cardiometabolic diseases; cellular signaling pathway; endocrine disruptor; metabolism; lipid

## 1. Introduction

Bisphenol A (BPA) is a well-known endocrine disruptor with estrogenic activity and estrogen receptor (ER) binding ability. Humans are exposed to BPA through several routes, such as air, river and marine water, and soil, but the major route of exposure is through BPA-contaminated foods [1,2]. BPA contamination occurs when food comes in contact with BPA-containing materials or containers, such as polycarbonate plastics and epoxy resin-based coatings for aluminum and steel cans, and BPA migrates from these surfaces into the food [2,3].

BPA is degraded or metabolized by many living organisms, microorganisms, plants, amphibians, fish, and mammals [4,5]. However, despite its susceptibility to being biodegraded or metabolized in living organisms, many endocrine-disruptive and toxic effects of BPA have been reported. As discussed in our previous review, BPA induces carcinogenesis, reproductive toxicity, abnormal inflammatory or immune responses, and developmental disorders of the brain or nervous system through various cellular signaling pathways [6].

Cardiovascular or cardiometabolic diseases, including diabetes, obesity, hypertension, acute kidney disease (AKD), chronic kidney disease (CKD), and heart failure, are the leading risk factors for global deaths. Recent evidence suggests positive associations between blood or urinary BPA levels and cardiovascular or cardiometabolic diseases. BPA has also been suggested to induce disorders or dysfunctions in the tissues associated with these diseases through various cell signaling pathways.

This review highlights the literature elucidating the relationships between BPA and various cardiovascular or cardiometabolic diseases and the potential mechanisms underlying BPA-mediated disorders or dysfunctions in the tissues associated with these diseases, such as blood vessels, skeletal muscle, adipose tissue, liver, pancreas, kidney, and heart. Furthermore, BPA-mediated disorders or dysfunctions and their related signaling pathways are summarized in figures within each section.

## 2. Relationship between Blood or Urinary BPA Levels and Cardiovascular or Cardiometabolic Diseases

Increased exposure to BPA in humans, mainly by the consumption of BPA-containing foods, is likely to increase the BPA levels in urine or blood [2,7,8]. Blood and urinary BPA levels are positively associated with an increased risk of cardiovascular or cardiometabolic diseases.

### 2.1. BPA Levels and Diabetes

Many global reports indicate significantly high correlations between urinary BPA levels and the risk of type 1 (T1D) or type 2 diabetes mellitus (T2D) onset in individuals from countries such as China [9,10], France [11], Iran [12], Korea [13], Mexico [14], Saudi Arabia [15], Pakistan [16], Thailand [17,18], and the US [19–22]. Both free BPA and total BPA concentrations were higher in urine samples obtained from diabetic patients than in those obtained from non-diabetic patients [13]. Increased concentrations of urinary BPA are also associated with an increased risk of diabetes among middle-aged but not older women [22]. Furthermore, an analysis of 3782 adults ($\geq$19 years) showed a significantly higher risk of diabetes mellitus in the highest quartiles of BPA than in the lowest quartiles [23]. Predialysis serum BPA levels were significantly higher in patients with diabetes than in those without diabetes [24]. In an analysis of 2336 adult women ($\geq$40 years), an inverse association was observed between urinary BPA and the glucose metabolic marker homeostasis model assessment-B (HOMA-B) in overweight or obese individuals than in those with normal weight. Urinary BPA levels were positively associated with fasting hyperglycemia and $\beta$-cell dysfunction in women but not in men [10]. In women, but not in men, higher urinary BPA levels are positively related to prediabetes, which is the earlier stage in the hyperglycemia continuum and is associated with an increased risk of developing diabetes [25]. In a study, BPA levels in the serum samples were positively correlated with senescence indicators such as GLB1, p16, p21, and p53; inflammatory markers such as IL6 and TNF-$\alpha$; and estrogen-related receptor $\gamma$. A negative correlation was reported with telomere length in patients with T2D [26].

In 620 pregnant women, high maternal urinary BPA concentrations led to a reduced risk of gestational diabetes mellitus (GDM) and slightly reduced birth weight and ponderal index [9]. Maternal serum BPA levels were also negatively associated with the risk of developing GDM [27]. In another study, serum BPA levels showed a positive relationship with blood glucose and insulin levels and HOMA-insulin resistance in the middle term of pregnancy; an increased BPA concentration tended to increase the risk ratio of GDM, although this was not statistically significant [28].

Interestingly, an inverse association between urinary BPA and insulin resistance was found in 107 healthy normal-weight children (58 girls) in the age range of 8.5–16.1 years. Children in the lowest (vs. highest) BPA tertile showed higher peak insulin levels in oral glucose tolerance tests; lower insulin sensitivity indices; higher levels of leptin, triglyceride (TG), and total cholesterol; lower aerobic fitness; and a tendency toward a higher fat mass index [19].

A direct association between urinary BPA levels and HOMA and an inverse association between urinary BPA and serum adiponectin levels were found in 141 obese children aged 4–16 years [29]. In an analysis of children and adolescents with T1D aged 3–25 years (n = 75) and age-matched participants (n = 113), the T1D group had significantly higher urinary BPA levels than those in the control group [18]. Additionally, an inverse relationship between

urinary BPA levels and birth weight was identified in 50 children with T1D aged 5–18 years, whereas no significant association was found between urinary BPA levels and T1D [30].

Glycosylated hemoglobin A1c (HbA1c) is a form of hemoglobin, higher levels of which can be indicative of diabetes. Several studies have reported a positive correlation between HbA1c and urinary BPA levels [12,16,31,32]. Urinary BPA levels were higher in diabetic participants than in non-diabetic participants and showed a positive association with serum risk factors for diabetes mellitus such as HbA1c, HOMA-insulin resistance, C-reactive protein (CRP), blood urea nitrogen (BUN), aspartate transaminase (AST), free fatty acids (FFAs), TGs, and malondialdehyde [16]. Furthermore, a positive association between urinary BPA quartiles and HbA1c levels was observed in men but not in women and children [32].

However, some studies have shown no association between urinary BPA concentrations and diabetes [33–36].

### 2.2. BPA Levels and Obesity

Urinary BPA levels are associated in a dose-responsive fashion with modestly greater weight gain in women from the US aged 30–55 years [37]. High urinary BPA concentrations also increase the risk of incident central obesity in Chinese women and individuals who are aged <60 years with normal weight, non-smokers, non-drinkers, and non-hypertensive [38]. In urine samples from Korean adults (2012–2017; n = 10,021), urinary BPA levels were significantly higher in obese than in non-obese adults and were significantly positively associated with the odds of obesity in both sexes, although more prominently in females than in males [39]. Furthermore, when urine samples of adults from the US (2013–2016; n = 1046) and Korea (2015–2017; n = 3268) were analyzed, urinary BPA levels were associated with decreased high-density lipoprotein (HDL) cholesterol levels, increased TG levels, and higher odds of obesity in both populations [40]. Additionally, analytical data for Canadian adults aged 18–79 years (n = 4733) showed that urinary BPA was positively associated with body mass index (BMI)-defined obesity but not with elevated waist circumference in the highest (vs. lowest) BPA quartile [41].

Urinary BPA concentrations are positively associated with the risk of general and abdominal obesity in boys more than in girls aged 6–17 years, suggesting a possible sex-related difference [42]. Among 63 prepubertal children, obese children with metabolic syndrome showed higher urinary BPA levels than obese children without metabolic syndrome, and both obese groups had considerably elevated levels of urinary BPA compared to those of the non-obese groups [43]. Data pertaining to BPA concentrations in urine samples of children aged 6–19 years revealed a decreasing trend in urinary BPA concentration from 2003 to 2014. Children with higher urinary BPA concentrations were associated with elevated odds for obesity from 2003 to 2008, whereas these associations were inconsistent from 2009 to 2014 [44]. Furthermore, an analysis of urine samples from 298 boys aged 9–11 years showed positive relationships between urinary BPA levels and higher BMI z-scores, higher waist-to-height ratios, and increased odds for overweight/obesity [45].

Interestingly, an analysis of mother–child pairs (mothers and their children aged 3–7 years; n = 430) showed that maternal urinary BPA concentration was positively associated with waist circumference in children aged 7 years [46]. BPA levels in urine samples collected from women at 6.3–15 weeks of gestation was correlated with the weight, height, waist/hip circumference, and subscapular/triceps skinfold thickness of their children aged 1.9–6.2 years. The results showed that gestational urinary BPA levels were positively associated with central adiposity in female children during early childhood [47].

However, no relationship has been observed between urinary BPA and obesity in children and adolescents. For example, no significant associations were found between urinary BPA levels and childhood obesity in Indian children aged 2–14 years [48]. Similarly, urinary BPA concentrations during childhood are associated with greater adiposity in children aged 8–12 years [49]. Furthermore, urinary BPA is not significantly associated with

general obesity, abdominal obesity, or any body mass outcome in children and adolescents aged 6–19 years [50].

### 2.3. BPA Levels and Hypertension

Several studies have suggested that urinary BPA concentrations are positively associated with blood pressure. For example, urinary BPA levels are associated with hypertension independent of traditional risk factors such as age, sex, race/ethnicity, smoking, BMI, diabetes mellitus, and total serum cholesterol levels [51]. In 521 participants aged ≥60 years, the diastolic blood pressure, systolic blood pressure (SBP), and mean heart rate increased, whereas the root mean square of successive differences in heart rate decreased with increasing urinary BPA concentrations [52,53]. Furthermore, in a cross-sectional study that included 1437 eligible participants without hypertension-related diseases, the individuals in the middle and high BPA exposure groups showed increased hypertension and higher SBP levels, with an inverted U-shaped dose–response relationship, compared with those in the control group [54]. In a case–control study with 439 pairs of hypertensive patients and matched control participants, BPA exposure was positively associated with the risk of hypertension. Moreover, the modified effects of ERα/β, catalase, and endothelial nitric oxide (eNOS) have been observed in the association between BPA exposure and the risk of hypertension [55]. Interestingly, when 60 participants were provided with the same beverage in two glass bottles, two cans, or one can and one glass bottle at a time, higher urinary BPA concentrations were detected in those who consumed canned beverages than in those who consumed glass-bottled beverages. SBP adjusted for daily variance was significantly increased after the consumption of two canned beverages than after the consumption of two glass-bottled beverages [52]. In a similar experiment, 60 participants were provided with the same beverage in two glass bottles, two cans, or one can and one glass bottle at a time. Decreased miR-30a-5p, miR-580-3p, miR-627-5p, and miR-671-3p levels and increased miR-636 and miR-1224-3p levels that could be attributed to BPA exposure were observed and were associated with high blood pressure [56]. Additionally, based on the serum BPA levels, the fourth quartile of women (mainly postmenopausal women) exhibited a higher risk of hypertension than those in the lowest quartile. However, no interaction was observed between serum BPA and estradiol levels [57].

Contrary to the results of these studies, Wang et al. suggested that urinary BPA concentrations are negatively associated with hypertension and early macrovascular diseases among middle-aged and elderly Chinese persons [58]. Moreover, urinary BPA levels were not associated with gestational hypertension (GH) or preeclampsia in 1909 pregnant women but were related to a decreased risk of GH in multiparous women [59].

### 2.4. BPA Levels and Kidney Diseases

AKD includes acute abnormalities in kidney function and structure and is associated with an increased risk of the development or progression of CKD and end-stage renal disease (ESRD) [60,61]. Analysis of adult participants in the US (2005–2016; n = 9008) revealed an association between urinary BPA concentrations and reduced estimated glomerular filtration rate (eGFR) and increased albumin-to-creatinine ratio (ACR) at the general exposure levels among adults [62]. In another study using samples collected from children and adolescents with CKD from the US and Canada (2005–2015; n = 618), urinary BPA levels were positively associated with urinary biomarkers of tubular injury such as kidney injury molecule-1 (KIM-1) and neutrophil gelatinase-associated lipocalin (NGAL) and urinary oxidative stress biomarkers such as 8-hydroxy-2′-deoxyguanosine (8-OHdG) and $F_2$-isoprostane; however, no association was found with eGFR, proteinuria, or blood pressure [63]. A cohort study that included children (1–17 years; n = 538) from the US with impaired kidney function showed that urinary BPA levels in children with CKD were consistently lower than those detected in healthy children; however, no association was observed between urinary BPA levels and blood pressure, proteinuria, or eGFR [64]. In healthy Korean adults (2015–2017; n = 1292), urinary BPA levels were negatively associated with eGFR in the total population [65]. Additionally, associations

between urinary BPA levels and urinary albumin creatinine ratio were found to be negative at RF-1 (estimated glomerular filtration rate (eGFR) > 90 mL/min/1.73 m$^2$), positive at RF-2 (60 ≤ eGFR ≤ 90 mL/min/1.73 m$^2$), negative at RF-3A (45 ≤ eGFR < 60 mL/min/1.73 m$^2$), and negative at RF-3B/4 (15 ≤ eGFR < 45 mL/min/1.73 m$^2$) [66].

Furthermore, several studies have reported a relationship between serum BPA concentration and AKD or CKD. For example, an analytic study of the middle-aged and elderly Chinese population (n = 1370) showed that serum BPA levels were negatively correlated with eGFR levels. Participants with high serum BPA levels had a significantly negative association with CKD compared with those with low BPA levels [67].

In serum samples collected from patients with CKD (n = 58), patients on dialysis therapy (n = 66), and healthy participants (n = 30), serum BPA levels were correlated with decreased eGFR in patients with CKD and healthy participants, and the levels were higher in patients undergoing hemodialysis than in those undergoing peritoneal dialysis [68]. Moreover, in a 6-year prospective study of 302 patients with primary hypertension [69] or 121 patients with T2D [70], serum BPA levels were negatively associated with annual changes in eGFR levels. Patients with high serum BPA levels showed a significantly higher risk of developing CKD than those with lower levels.

Increased serum BPA levels have been observed in patients undergoing hemodialysis. This increase is associated with the hemodialysis membrane [71–73]. Higher serum BPA levels have been detected in patients using polysulfone membranes than in those using cellulose [73] or polynephron membranes [71,72]. Moreover, BPA was released from polysulfone [68], polyamide [68], and polycarbonate membranes [74], but no BPA release was detected from polypropylene membranes [74].

### 2.5. BPA Levels and Cardiovascular Diseases

Growing evidence indicates that urinary or serum BPA concentrations are positively correlated with the risk of cardiovascular diseases (CVDs) such as myocardial infarction (MI), stroke, and coronary artery disease (CAD; also called coronary heart disease). High urinary BPA concentrations are significantly associated with an increased risk of CVD mortality [75].

In a representative sample of individuals aged ≥20 years (2003–2014; n = 9139), urinary BPA concentration was associated with an increased prevalence of congestive heart failure, CAD, angina pectoris, MI, stroke, and total CVD. These associations were more evident in males than in females [76]. Similarly, a cross-sectional study of 8164 individuals (2003–2012) showed a positive relationship between BPA levels and the risk of CVD [77]. Moreover, in 591 patients with severe CAD (n = 385), intermediate disease (n = 86), and normal coronary arteries (n = 120), urinary BPA concentration was significantly higher in those with severe CAD and nearly significant for intermediate disease compared with that in those with normal coronary arteries. No significant difference in urinary BPA levels was observed between patients with severe CAD (requiring surgery) and those in the remaining groups combined [78]. Analytical data on participants aged 40–74 years (861 controls and 758 cases of incident CAD) showed positive associations between urinary BPA levels and incident CAD [79]. In a cross-sectional analysis of 574 adults, the high urinary BPA tertile showed higher levels of low-density lipoprotein (LDL) cholesterol and lower levels of HDL cholesterol than in the low BPA tertile [80].

In a study using plasma samples from 90 young adults aged 20–45 years, the high plasma BPA group showed higher BMI, blood pressure, plasma CRP level (an inflammation biomarker of CVD), and gene expression of ERβ and inhibitor of NF-κB kinase (IKK)β compared with that in the low BPA group. Participants with potential CVD development symptoms had increased plasma BPA and CRP levels and IKKβ gene expression. These results suggest that BPA-mediated inflammation increases the risk of developing CVD symptoms in young adults [81]. Furthermore, an analysis of 886 young participants aged 12–30 years showed a relationship between serum BPA levels and carotid artery intima-media thickness, which is a surrogate marker of subclinical atherosclerosis. These results

suggest that BPA exposure increases endothelial dysfunction and subclinical atherosclerosis in young populations [82]. In an investigation of patients with dilated cardiomyopathy (DCM) (n = 88) and age- and sex-matched healthy control participants (n = 88), significantly higher serum BPA levels were detected in patients with DCM than in the healthy participants [83]. In a cross-sectional study (1016 participants aged 70 years), high serum BPA levels were associated with plaque echogenicity (grayscale median ((GSM)), intima media thickness (IMT), and GSM (IM-GSM), indicating that BPA exposure leads to atherosclerotic plaque enhancement [84].

Conversely, another study reported that urinary BPA was not significantly associated with the risk of CAD, heart attack, or diabetes [35].

### 2.6. BPA Levels and Liver Diseases

Urinary BPA levels are positively associated with the risk of non-alcoholic fatty liver disease (NAFLD) in adults from Korea (n = 3476) [85] and the US (n = 7605) [86] and Hispanic adolescents (n = 944) [87]. Positive relationships were also observed between urinary BPA concentrations and increased serum levels of liver enzymes such as AST, alanine aminotransferase (ALT), and γ-glutamyl transferase and increased abnormal liver function in older Korean adults (n = 560) [88], children (n = 164) [89], and adults from the US (n = 1455) [20]. Furthermore, serum BPA levels showed negative effects on hepatic function in Chinese children (n = 1006) [90].

## 3. Effects of BPA on Cardiovascular or Cardiometabolic Organs

### 3.1. Effects of BPA on Skeletal Muscles

The skeletal muscles play an important role in glucose homeostasis and insulin resistance. Several findings have revealed BPA-related dysfunction in glucose homeostasis and insulin resistance in skeletal muscles (Figure 1). C2C12 myotubes treated with BPA (1000 ng/mL) for 24 h showed increased GLUT4 mRNA levels but reduced insulin receptor (IR) levels, whereas the insulin receptor substrate-1 (IRS-1) and AMP-activated protein kinase (AMPK) mRNA levels were unchanged [91]. Furthermore, BPA (100 μM)-treated L6 myotubes showed decreased metabolic activity during the last 24 h of differentiation but increased insulin-stimulated glucose uptake, phosphorylation of the insulin signaling protein Akt, and glycolysis [92]. BPA treatment (100 μg/kg/day) during gestational days (GD) 9–16 reduced insulin-stimulated Akt phosphorylation in the skeletal muscle and liver of pregnant mice [93].

Furthermore, oral BPA treatment decreased the protein levels of insulin signaling-related molecules such as IR, Akt, and Akt substrate of 160 kDa (AS160; also known as TBC1D4), as well as glucose transporter type 4 (GLUT4) translocation, resulting in reduced glucose uptake and oxidation in the gastrocnemius muscle of rats [94,95]. Downregulation of IRS-1 protein was observed in the skeletal muscle of rats orally treated with 400 mg/kg/day of BPA for 30 days [95], but subcutaneous injection of 100 μg/kg of BPA for 8 days in mice led to its upregulation in skeletal muscles [96]. In high-fat diet (HFD)-fed mice, long-term oral exposure to BPA (50 μg/kg/day) for 12 weeks reduced Akt phosphorylation at Thr308 and glycogen synthase kinase 3β phosphorylation at Ser9 in skeletal muscles, thereby inducing glucose intolerance and insulin resistance in growing mice [97]. BPA-treated mice (subcutaneous injection of 100 μg/kg of BPA for 8 days) also showed reduced insulin-stimulated Akt phosphorylation at Thr308, insulin-stimulated tyrosine phosphorylation of the insulin receptor β subunit, and downregulation of the MAPK signaling pathway (extracellular signal-regulated kinase (ERK) 1/2) in skeletal muscles [96]. Rat dams exposed to BPA through the drinking water 2 weeks prior to mating and through pregnancy and lactation showed higher IR-β levels in male offspring and lower GLUT4 levels in the skeletal muscles in both male and female offspring at 10 months of age than control rats [98].

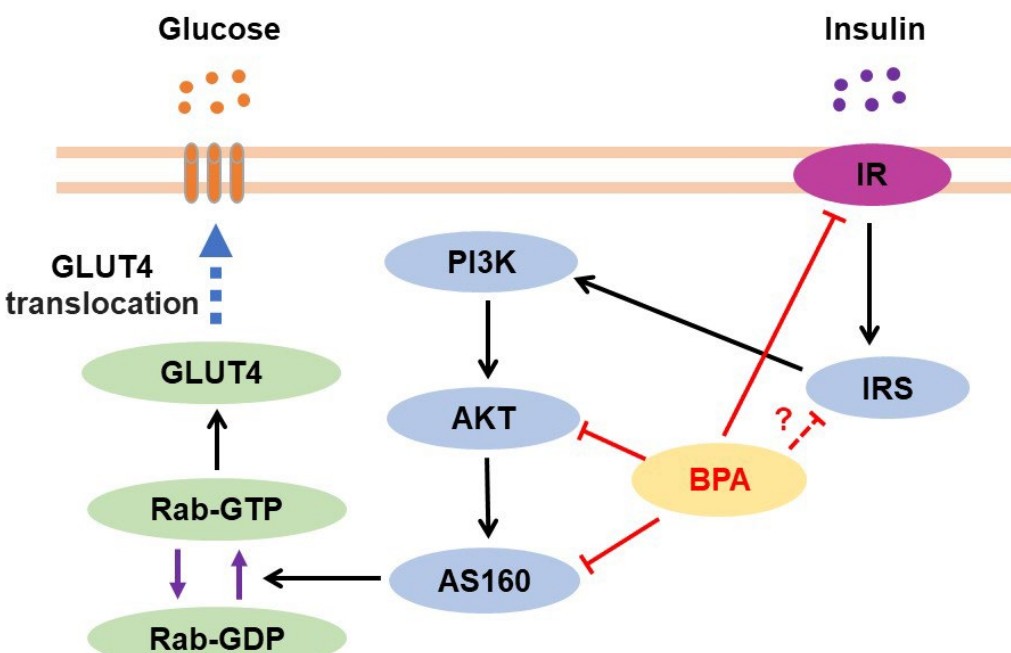

**Figure 1.** Potential effects of bisphenol A (BPA) on the insulin-signaling pathway and glucose transport in skeletal muscle. Insulin-induced insulin receptor (IR) activation stimulates the phosphorylation of AKT and its substrate AS160. Activated AS160 elevates the GTP-bound forms of Rabs, which increases GLUT4 translocation, resulting in increased glucose uptake in skeletal muscle. However, BPA decreases glucose uptake by inhibiting the AKT/AS160/GLUT4 axis. Black arrows and red bars indicate activation and inhibition cascades, respectively. AS160; Akt substrate of 160 kDa (also known as TBC1D4); GLUT4, glucose transporter type 4; IR, insulin receptor; IRS-1, insulin receptor substrate-1; PI3K; phosphatidylinositol 3-kinase.

These results indicate that BPA disrupts glucose homeostasis in the skeletal muscles by inhibiting the Akt/AS160/GLUT4 axis. Furthermore, insulin-stimulated AS160 phosphorylation was significantly reduced in the skeletal muscles of patients with T2D compared with control participants, whereas AS160 protein expression was similar in both groups [99]. Therefore, BPA exposure in patients with T2D can accelerate the potential adverse effects on glucose homeostasis in skeletal muscles.

$Ca^{2+}$ plays a critical role in excitation–contraction coupling in the skeletal muscle. Perfusion of 30 μM BPA to skeletal myotubes impaired the excitation–contraction coupling of myotubes through rapid $Ca^{2+}$ transient loss caused by cell hyperpolarization, which alters cellular excitability [100]. However, very few studies have demonstrated the adverse effects of BPA on the $Ca^{2+}$ signaling pathway in skeletal muscles. Therefore, further studies are needed to better understand the BPA-induced alteration of the $Ca^{2+}$ signaling pathway in skeletal muscles, especially the $Ca^{2+}$/CaMKK/AMPK axis [101].

## 3.2. Effects of BPA on Adipocytes and Adipose Tissues

### 3.2.1. BPA-Mediated Adipogenesis

Adipocytes (also known as adipose cells or fatty cells) are the primary cells and major energy storage sites in adipose tissues (fat tissues or fatty tissues). BPA influences the adipogenesis of adipocytes (Figure 2). During the preadipocyte differentiation process, enhanced adipogenesis was observed in females in human adipose-derived stem cells (hASCs) treated with 0.1 μM [102] or 0.1–1 μM BPA [103], but decreased adipogenesis was observed in cells affected with Simpson–Golabi–Behmel syndrome (SGBS) after BPA treatment (0.01–10 μM) [104]. Cohen et al. used hASCs purchased from Lifeline Cell Technology (FC-0062; Frederick, MD, USA) [102], but Ohlstein et al. used hASCs obtained from the subcutaneous abdominal adipose tissues of three Caucasian females with a BMI

below 25 (average age of 34.6 ± 8.4 and an average BMI of 22.2 ± 1.1) [103]. In contrast, SGBS cells were derived from the stromal cell fraction of the subcutaneous adipose tissue of an infant with SGBS and showed high similarity to differentiated human primary preadipocytes [105]. These cells showed different viabilities in the presence of BPA. Cohen et al. observed increased cell death and decreased cell viability in hASCs treated with ≥1 μM BPA for 24 h [102]. When preadipocytes were differentiated, cytotoxicity was observed in hASCs following treatment with 10 μM of BPA for 21 days, but no change in cell viability was observed in the SGBS cells treated with BPA concentrations of 0.01–10 μM for 12 days [104].

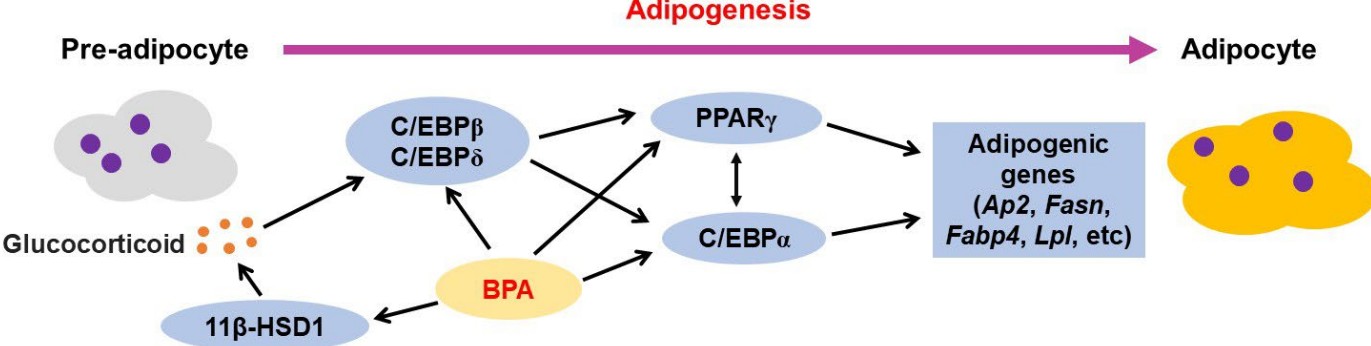

**Figure 2.** Potential effects of BPA on the adipogenesis of adipocytes. BPA increases the expression of adipogenic genes by directly stimulating PPARγ and C/EBPα or by stimulating their upstream regulators C/EBPβ and C/EBPδ. Black arrows indicate activation cascades. AS160, Akt substrate of 160 kDa; PPAR, peroxisome proliferator-activated receptor; C/EBP, CCAAT/enhancer-binding protein.

BPA exposure (0.1–10 nM) increased the mRNA levels of *C/ebpa*, *C/ebpb*, *Lpl*, and *Fasn* but decreased the protein levels of GLUT4 and FABP4 after 6 days during the differentiation of hASCs [106]. Furthermore, the early stage adipogenic gene *Dlk*, mid-stage adipogenic gene *C/ebpa*, late-stage adipogenic genes *Igf1* and *Lpl*, and master transcriptional regulator of adipogenesis *Pparg* expression increased in the hASCs after 7 days following BPA treatment (1 μM), whereas no changes were noticed in the expression of other genes such as *Srebp1c*, *aP2*, and *C/ebpb* [103]. In SGBS cells, *Pparg*, *Fabp4*, *Lpl*, and apolipoprotein E (*Apoe*) are downregulated by BPA (1 μM) [104].

BPA (1–1000 nM) in the presence of the glucocorticoid dexamethasone (0.5 nM), a hormone required for the differentiation of 3T3-L1 cells, increased aP2 protein levels on day 8, whereas aP2 levels increased in the cells treated with 10 nM BPA alone. These results indicate that BPA potentiates the ability of dexamethasone to enhance the differentiation of 3T3-L1 cells [107]. Moreover, BPA treatment (50–80 μM) promotes adipocyte differentiation of 3T3-L1 preadipocytes [108].

3T3-L1 adipocytes differentiated with 1 nM BPA showed increased *Pparg* and *Fabp4/Ap2* mRNA levels on day 8 and increased *C/ebpa* mRNA level on day 4. PPARγ protein levels increased significantly both on days 4 and 8 of adipogenesis, whereas that of FABP4/AP2 only increased on day 8 [109]. Moreover, BPA (1 nM)-exposed 3T3-L1 preadipocytes showed significantly increased *Pparg* expression and enhanced lipid droplet accumulation on day 4 of differentiation, but this was not observed in terminally differentiated adipocytes (day 8) [110]. However, 3T3-L1 adipocytes differentiated with 100 nM BPA showed decreased mRNA expression for several adipocyte markers, such as *Pparg Fabp4* and *Fasn*, between days 8 and 9 [111]. In another study, BPA (25 μM)-treated 3T3-L1 cells showed increased aP2 levels through the activity of C/EBPδ and increased glucocorticoid receptor levels through PPARγ and C/EBPα [107].

When Sprague Dawley rats received purified drinking water containing BPA (5 mg/L) 2 weeks prior to mating and throughout pregnancy and lactation, the 1-day-old male offspring showed increased expression of the protein PPARγ, but not of the lipogenic factor sterol regulatory element binding protein 1 (SREBP1; also known as sterol regulatory

element-binding transcription factor 1). Increased expression of both C/EBPα and SREBP1 was observed at 3 weeks. Markedly increased adipose tissue mass and hypertrophic adipocytes were observed in male rats at 3 weeks of age; however, no alteration in adiposity was observed in females [112]. Dunder et al. investigated the levels of stearoyl-CoA desaturase 1 (SCD-1; a key regulator of lipid metabolism) in offspring after pregnant F344 rats were exposed to BPA (0.5 or 50 µg/kg/day) via drinking water from GD 3.5 until postnatal day (PND) 22. Increased SCD-16 and SCD-18 indices in inguinal white adipose tissue (iWAT) TG and plasma cholesterol esters were found in 5-week-old male offspring exposed to BPA (0.5 µg/kg) but not in males treated with BPA (50 µg/kg/day). Moreover, BPA exposure (0.5 µg/kg) altered fatty acid (FA) composition in male offspring by decreasing the levels of the essential polyunsaturated FA linoleic acid (18:2n-6) in iWAT and liver TG. In contrast, no alterations were noticed in the SCD-1 levels and FA composition in 5-week-old female offspring exposed to maternal BPA (0.5 and 50 µg/kg) [113].

Dosing pregnant CD-1 mice with BPA (5 or 500 µg/kg/day) on GD 9–18 led to significantly increased gonadal fat pad weight in 19-week-old male offspring exposed to both doses of BPA; however, the same effect was elicited only by exposure to 500 µg/kg BPA in a 19-week-old female. Compared with that of the negative controls, the 19-week-old male offspring exposed to both doses of BPA also showed significantly increased gonadal fat cells and adipocyte volume, but the male offspring were exposed to maternal BPA at the dosage 5000 µg/kg/day [114,115]. Stronger mRNA expression of *Fggy* (an obesity-relevant gene) was found in gonadal fat isolated from 19-week-old males but not in female gonadal fat. *Fggy* mRNA expression is significantly correlated with increased body weight (BW) and gonadal fat weight in males [115]. Furthermore, 5-week-old male and female C57BL/6J mice exposed to BPA (5–5000 µg/kg/day) orally for 30 days showed increased BW and fat mass when fed a chow diet but not an HFD. Expression of adipogenic genes such as *C/ebpa*, *Pparg*, and *Ap2* was also increased in the iWAT of male mice fed a chow diet following BPA exposure. BPA treatment (50 µM) activates the differentiation of white adipocyte progenitors from the stromal vascular fraction through GR and increases *C/ebpa*, *Pparg*, and *Ap2* mRNA expression [116].

In addition, the enzyme 11β-hydroxysteroid dehydrogenase type 1 (11β-HSD1) promotes adipogenesis in adipose tissues by converting the inactive hormone cortisone to the active hormone cortisol and activating glucocorticoid production [117]. BPA (10 nM) induced the mRNA expression and enzymatic activity of 11β-HSD1, mRNA expression PPAR-γ and *LPL*, and lipid accumulation in omental adipose tissues and visceral adipocytes obtained from children [118].

ERα in adipose tissue and adipocytes protects against adiposity, inflammation, glucose intolerance, and insulin resistance in males and females. In the absence of ERα, ERβ plays a protective role in the adipose tissue and adipocytes [119,120]. Ohlstein et al. showed that BPA addition (1 µM) significantly increased the mRNA expression of *ERα* and *ERβ* on day 7 and *ERα* alone on day 14 of differentiation in hASCs. Despite increased ER mRNA expression, enhanced adipogenesis was observed in cells exposed to 1 µM BPA for 21 days [103]. Nevertheless, another study reported that BPA exposure (0.1–10 nM) decreased the protein levels of ERβ on day 14 during the differentiation of hASCs [106]. In 3T3-L1 cells, BPA (50–80 µM) was associated with the activation of ERα in undifferentiated 3T3-L1 cells and ERβ in differentiated 3T3-L1 cells [108]. Furthermore, BPA addition (10 and 100 nM) to mature adipocytes derived from preadipocytes of children increased the mRNA expression of *ERα* at 24 and 48 h, whereas no changes were observed in the *ERβ* mRNA levels [121]. Mature adipocytes from mammary subcutaneous AT showed no changes in ERα and ERβ levels after 24 and 48 h of BPA exposure (0.1 nM) [122]. In human abdominal subcutaneous AT, the expression of *ERα* and *ERβ* was also unaffected by BPA exposure (1–1000 nM) for 24 h [123]. Therefore, further studies are needed to clarify whether BPA-mediated adipogenesis is dependent or non-dependent on the ER pathway.

3.2.2. BPA-Mediated Inflammatory Response and Adipokine Dysregulation

Adipose tissue inflammation is a risk factor for increased insulin resistance, obesity, and systemic inflammation. Inflammatory responses in adipose tissues induce the dysregulation of secreted adipokines such as leptin, resistin, and adiponectin (also known as ADIPOQ); induce the production of inflammatory cytokines such as interleukin (IL)-1 and tumor necrosis factor (TNF)-α; and induce the infiltration of immune cells such as monocytes and macrophages [124]. BPA is associated with inflammatory responses and adipokine dysregulation in adipocytes and adipose tissue (Figure 3).

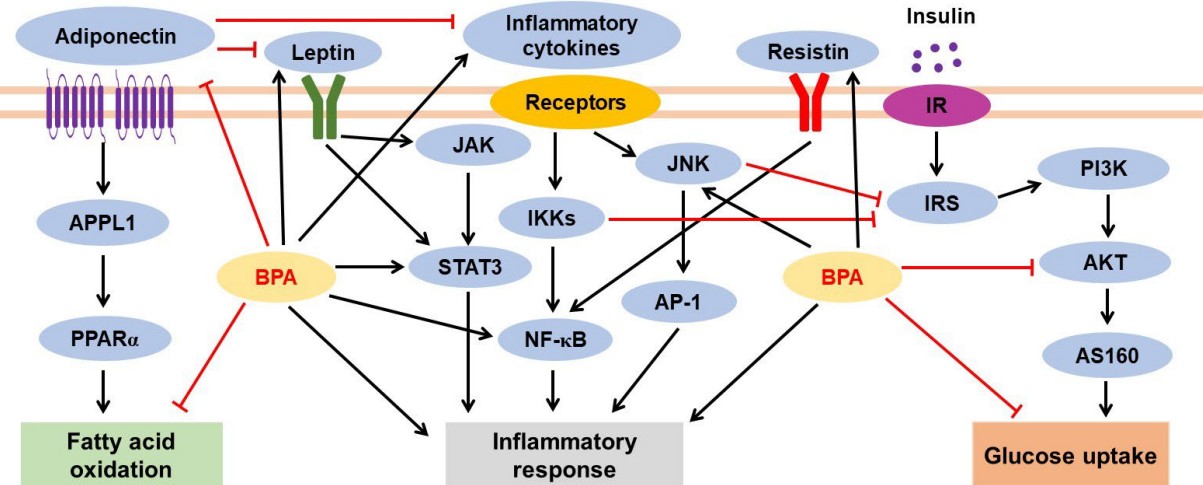

**Figure 3.** Effects of BPA on inflammatory response and adipokine dysregulation in adipocytes. Adipocytes can secret both pro-inflammatory (e.g., leptin and resistin) and anti-inflammatory adipokines (e.g., adiponectin) as regulators of inflammatory responses and metabolic function. Adiponectin participates in fatty acid oxidation via the APPL1/PPARα axis. BPA increases pro-inflammatory adipokine production but decreases anti-inflammatory adipokine secretion. Inflammatory cytokines (e.g., IL-1 and TNF-α)-mediated inflammatory responses are stimulated via JAK/STAT3, IKKs/NF-κB, and JNK/AP-1 cascades. BPA activates these inflammatory cascades. Black arrows and red bars indicate activation and inhibition cascades, respectively. AP-1, activator protein-1; APPL1, adaptor protein containing pleckstrin homology domain, phosphotyrosine binding domain, and leucine zipper motif 1; AS160, Akt substrate of 160 kDa; IKK, IκB kinase; IR, insulin receptor; IRS-1, insulin receptor substrate-1; JAK, Janus kinase; JNK, c-Jun NH2-terminal kinase; NF-κB, nuclear factor-κB; PI3K, phosphatidylinositol 3-kinase; PPAR, peroxisome proliferator-activated receptor; STAT, signal transducer and activator of transcription.

Adipocytes participate in the dysregulation of adipokine secretion and inflammatory cytokine production. During the differentiation of human preadipocytes (SGBS cells), BPA exposure (1 μM) for 12 days upregulated monocyte chemoattractant protein-1 (also known as CCL2), which is a member of the C-C chemokine family. Leptin levels were increased at 10 μM BPA, but adiponectin, adipsin, and leptin levels showed no change at 1 μM BPA exposure [104]. BPA treatment (0.1 nM) for 24 h also increased the mRNA expression of *IL8*, *MCP1A*, and *IL6* in human mature adipocytes obtained from subcutaneous mammary adipose tissue via G protein-coupled estrogen receptor 30 (GPR30) [122]. Incubating human adipocytes derived from subcutaneous adipose tissue and differentiated 3T3-L1 cells with 1 nM BPA for 24 h activated the JNK, STAT3, and NF-kB pathways, inducing the release of IL-6 and interferon (IFN)-γ. However, a decrease in leptin mRNA levels was also observed in BPA (1 nM)-treated 3T3-L1 adipocytes [125]. Similarly, an increase in *Il6* and *Ifng* expression was observed in 3T3-L1 adipocytes differentiated with 1 nM BPA on day 8 of adipogenesis [109]. Moreover, 24 h treatment with 10 nM BPA enhanced the production of the inflammatory cytokines IL-1β, IL-18, and CCL20 in mature adipocytes [121]. Resistin expression was upregulated, but adiponectin expression was downregulated 24 h after BPA

treatment (10 and 100 nM) of mature adipocytes from eight normal prepubertal children [29]. However, human subcutaneous adipose tissue treated with BPA for 24 h showed reduced expression of pro-inflammatory cytokine genes (*IL6* at 10 nM and 10 μM, *IL1B* at 10 nM, and *TNFA* at 1 nM) and adipokines (*ADIPOQ* and *FABP4* at 10 nM); however, the expression of these genes was not affected after 72 h of BPA treatment [123].

In in vivo studies, a chow diet with BPA (5000 μg/kg/day) promoted the expression of inflammatory genes such as *Il6*, *Tnfa*, *Il1b*, *Ifng*, and *iNos2* in the iWAT of male mice. Plasma leptin and resistin levels also increased in both males and females fed a chow diet with BPA, but not in both sexes exposed to BPA under HFD conditions [116]. In male offspring exposed to maternal BPA (5000 μg/kg/day), increased serum leptin and insulin levels but decreased adiponectin levels were observed [114]. Gestational BPA treatment (0.5 mg/kg/day; from day 30 to 100) through daily subcutaneous injection decreased *ADIPOQ* expression in the adipose tissue of BPA-exposed fetal male and female sheep [126].

In a human study, plasma BPA and pro-inflammatory cytokine levels were significantly higher when the waist circumference of the study participants was >102 cm than when ≤102 cm. A positive correlation has been reported between plasma BPA and pro-inflammatory cytokine levels (IL-6 and TNF-α) [127]. Urinary BPA and adiponectin levels showed a strong inverse relationship in 74 obese children [29]. In a different study, however, serum levels of BPA were positively associated with adiponectin and leptin but negatively with ghrelin; this association was stronger in women than in men, whereas BPA levels and indices of fat mass or fat distribution showed no significant relationships [128].

Despite several efforts, further studies are needed to clarify the bigger picture regarding BPA-induced adipokine dysregulation and BPA-mediated inflammatory signaling pathways (e.g., AMPK and MAPK signaling pathways) in adipocytes and adipose tissue. Furthermore, whether BPA treatment can stimulate the infiltration of inflammatory immune cells such as monocytes and macrophages into adipose tissues is yet to be reported, although a study has shown that BPA (10 nM) increased the self-renewal of adipose tissue macrophages via ERK activation [129].

### 3.2.3. BPA-Induced Insulin Resistance

BPA inhibited insulin-stimulated glucose uptake in adipocytes (Figure 3). After incubation with insulin, decreased insulin sensitivity was observed in SGBS adipocytes differentiated with 1 μM BPA, mainly by reducing the pAkt/Akt ratio [104]. Insulin-stimulated glucose uptake was reduced in adipocytes isolated from adipose tissue treated with BPA (10 nM and 10 μM) for 24 h, whereas no BPA-dependent changes were found in GLUT4 and Akt protein levels and phosphorylated Akt levels [123]. Furthermore, insulin-stimulated glucose utilization was reduced in differentiated 3T3-L1 cells treated with BPA (1 nM). After insulin stimulation, Akt phosphorylation was significantly lower in undifferentiated (day 0) and differentiated 3T3-L1 cells (day 8) treated with BPA than in untreated cells, whereas ERK1/2 phosphorylation was observed only on day 8 [109]. Mature 3T3-L1 adipocytes differentiated in the presence of 1 nM BPA showed an approximately 25% reduction in insulin-stimulated glucose uptake. This was associated with a significant decrease in Akt phosphorylation, but no other changes were observed in the insulin signaling pathway, including the pIRS1/IRS1 ratio [111]. In addition, BPA treatment (1 nM) for 24 and 48 h followed by insulin stimulation for 10 min led to a time-dependent decrease in insulin-stimulated IR tyrosine kinase, Akt, and ERK1/2 phosphorylation in human adipocytes derived from subcutaneous adipose tissue and differentiated 3T3-L1 cells, whereas no BPA-dependent change was observed in IR, Akt, and ERK1/2 total protein levels. Interestingly, treatment with the JNK inhibitor SP600125 (20 μM) enhanced the effects of insulin on IR, Akt, and ERK1/2 phosphorylation in BPA-stimulated adipocytes by inhibiting the JNK/STAT3 cascade, indicating that inhibition of the inflammatory response in adipocytes can rescue insulin sensitivity [125]. In 3T3-L1 adipocytes treated with 80 μM BPA for 2–24 h, the levels of pIRS-1 and pAkt proteins were downregulated, but IRS-1 and

Akt protein levels did not change. BPA treatment for 6–24 h increased the secreted protein levels of the suppressor of cytokine signaling 3, which is a negative regulator of insulin signaling and is associated with insulin resistance [130].

Male mice offspring exposed to BPA at doses of 5–5000 µg/kg/day, but not 50,000 µg/kg/day, showed impaired glucose tolerance, which is measured as the ability to return the blood glucose levels to baseline after injection of glucose. The impaired ability to lower blood glucose levels in response to insulin injection was significant in males exposed to 5 and 5000 µg/kg/day BPA but not to 50 and 500 µg/kg/day BPA, compared with the controls [114]. Furthermore, daily subcutaneous administration of BPA (5 mg/kg/day) during GD 30–90 reduced plasma levels of the low-molecular-weight but not of high-molecular-weight forms of adiponectin in 21-month-old female offspring [131].

### 3.2.4. BPA Levels in Adipose Tissues

Very low concentrations of BPA seem to accumulate in adipose tissue. For example, after intravenous injection of BPA (100 µg/kg/BW), the maximal level of unconjugated BPA in mouse adipose tissue was observed at 0.25 h, but <0.01% of the administered dose remained in the adipose tissue after 24 h [132]. However, BPA can accumulate in adipose tissues due to its lipophilic and bioaccumulation properties [4,133]. In fact, relatively higher levels of BPA are detected in human adipose tissue than in plasma, liver, and brain tissues. For example, the BPA levels ranged from <0.8 to 20.9 ng/g wet weight (median concentration of 5.56 ng/g) in human adipose tissue and from 0.784 to 4.97 ng/mL (median 1.77 ng/mL) in plasma [134]. In another study, BPA concentrations ranged from 1.80 to 12.01 ng/g in the adipose tissue (mean, 5.83 ng/g) [135]. Furthermore, BPA was detected in adipose tissue at a concentration range of 1.12–12.28 ng/g (mean 3.78 ng/g), followed by liver at a concentration range of 0.90–2.77 ng/g (mean, 1.48 ng/g) and brain at a concentration range of <0.4–2.36 ng/g (mean, 0.91 ng/g) [136]. A recent study reported that BPA levels are higher in urine and breast adipose tissue of patients with cancer than in noncancerous tissues, and urinary BPA levels are positively correlated with breast adipose tissue BPA levels in patients with breast cancer [137].

These results indicate that BPA accumulates in adipose tissues at a higher level than in other organs and induces long-term abnormal effects on adipocytes and adipose tissues, such as increased adipogenesis, inflammatory responses, and risk of breast cancer incidence.

### 3.3. Effects of BPA on Pancreatic Tissues

Pancreatic islets (also known as islets of Langerhans) contain five hormone-producing cells: α-cells (glucagon), β-cells (insulin), δ-cells (somatostatin), γ-cells (also known as F-cells or PP cells; pancreatic polypeptide), and ε-cells (ghrelin) [138]. BPA participates in the dysfunction and abnormal regulation of pancreatic cells (Figure 4). BPA exposure (1 nM) suppressed low-glucose-induced intracellular $Ca^{2+}$ oscillations in α-cells isolated from mouse islets [139]. Under both low- and high-glucose conditions, BPA at concentrations of 0.1 and 1 µg/mL significantly stimulated insulin secretion and increased the expression of Hsp70 in beta TC-6 cells [140]. BPA-exposed INS-1 pancreatic β-cells showed increased DNA strand breaks, greater DNA migration from the nucleus to the comet tail, higher expression of DNA damage-associated proteins such as p53 and p-Chk2 (T68), enhanced intracellular reactive oxygen species (ROS) levels, and reduced intracellular glutathione levels [141]. In INS-1Eβ-cells, insulin secretion increased after 2 h of simultaneous exposure to BPA and glucose, but longer BPA exposures (24–72 h) showed no consistent effects on glucose-induced insulin secretion [142]. Furthermore, impaired mitochondrial function and induced swelling of mitochondria with loss of distinct cristae structures within the membrane were observed in β-cells isolated from rat islets after treatment with BPA (25 µg/L) [143].

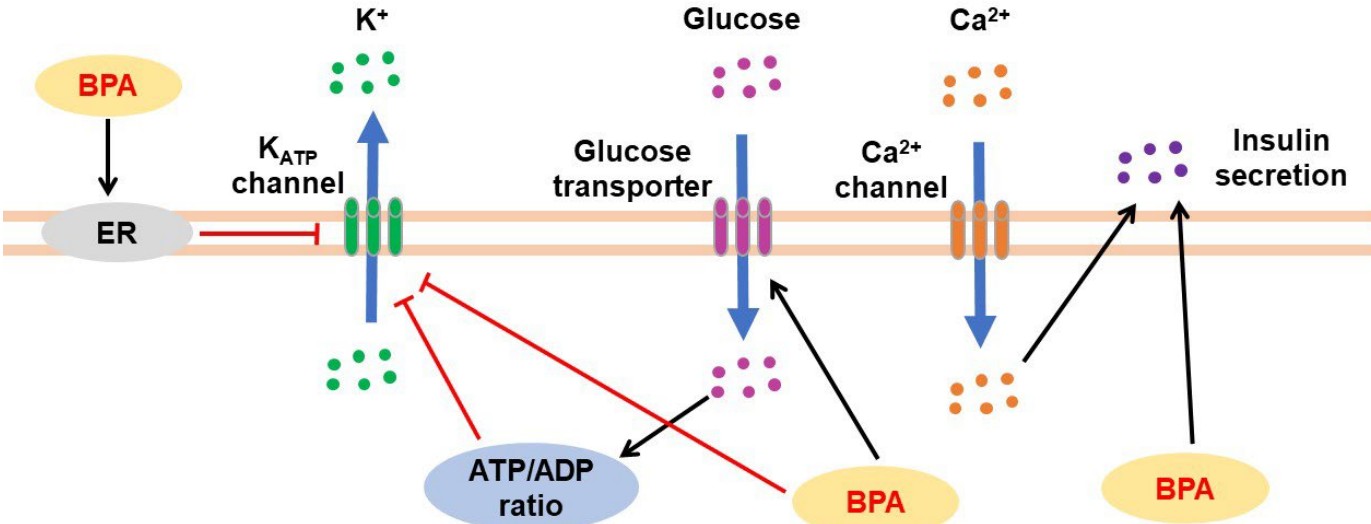

**Figure 4.** Low blood glucose level elicits the opening of the $K_{ATP}$ channel but the closure of the $Ca^{2+}$ channel (Cav2.3 ion channel) in pancreatic β-cells. However, the opposite functional response is elicited by high blood glucose levels. In a dose-dependent manner, BPA stimulates insulin secretion by the pancreatic β-cells and increases insulin content. Black arrows and red bars indicate activation and inhibition cascades, respectively. ADP, adenosine diphosphate; ATP, adenosine triphosphate; ER, estrogen receptor.

In addition to in vitro experiments, BPA-mediated pancreatic dysfunction has been observed in several in vivo studies. In rats exposed to BPA (0.5 or 50 μg/kg/day) via drinking water from GD 3.5 until PND 22, insulin secretion was enhanced in islets isolated from 5- and 52-week-old female and male offspring and from dams treated with 0.5 μg/kg/day BPA; however, the levels were reduced after treatment with 50 μg/kg/day BPA [144]. After treatment with low (10 μg/kg/day) or high (10 mg/kg/day) BPA doses 2 weeks prior to mating until weaning, high BPA exposure impaired mitochondrial function in F1 and F2 adult offspring. However, low BPA exposure reduced β-cell mass and increased β-cell death that persisted in the F2 generation [145]. After BPA treatment (100 μg/kg/day) during GD 9–16, 6-month-old male offspring from mothers treated with 100 μg/kg/day of BPA showed reduced glucose tolerance, increased insulin resistance, and altered blood parameters compared with those in offspring of untreated dams. Altered $Ca^{2+}$ signaling and insulin secretion have been observed in the islets of male offspring [93]. Furthermore, BPA exposure in pregnant mice led to an increase in pancreatic β-cell mass and proliferation and a decrease in β-cell apoptosis in offspring at PND 0, PND 21, and PND 30; equal or decreased β-cell mass was observed at PND 120 [146].

BPA-induced pancreatic dysfunction during the developmental period may be associated with altered DNA methylation of insulin-like growth factor-2 (*Igf2*) [145,147,148] and altered expression of *Pdx1* (a key regulator of insulin transcription) [149]. Moreover, BPA exposure during pregnancy alters the α:β-cell ratio in islets by increasing the number of glucagon-expressing islet-cell clusters in the fetal pancreas [150].

BPA regulates insulin secretion in pancreatic β-cells via an ER-mediated pathway. BPA treatment (1 nM) decreased $K_{ATP}$ channel activity in wild-type mice but increased glucose-induced intracellular $Ca^{2+}$ signals and insulin release in β-cells; these effects were not observed in the cells from $ERβ^{-/-}$ mice. Although a rapid reduction in $K_{ATP}$ channel activity and insulinotropic effect was also found in human β-cells and islets, the effects exerted by BPA were more pronounced in human islets than in mouse islets at the same concentration (1 nM) [151]. Furthermore, in pancreatic β-cells isolated from wildtype and $ERβ^{-/-}$ mice, low BPA doses (0.1–1 nM) decreased plasma membrane $Ca^{2+}$ currents after the downregulation of Cav2.3 ion channel expression via ERβ. However, high doses

(0.1–1 μM) decreased $Ca^{2+}$ currents through an ERβ-mediated mechanism and simultaneously increased the ERα-mediated $Ca^{2+}$ currents through a phosphatidylinositol 3-kinase (PI3K)-dependent pathway [152]. Additionally, BPA treatment reduced the activity of voltage-activated $K^+$ channel subunits and currents and large-conductance $Ca^{2+}$-activated $K^+$ channels in β-cells from wildtype mice but not in β-cells from $ERβ^{-/-}$ mice [153]. BPA exposure also upregulated insulin levels in β-cells via ERα activation through the ERK1/2 pathway [154].

Furthermore, BPA influences pancreatic β-cell division and mass via ERs. BPA exposure (10 μg/kg/day BPA) during days 9–16 of pregnancy increased pancreatic β-cell proliferation and mass in wildtype male offspring but not in the offspring from ERβ knockout mice, indicating that BPA induces ERβ-dependent upregulation of pancreatic β-cell division and mass during pregnancy [155]. In a recent study using ovariectomized mice, long-term exposure to BPA (1 μg/mL in drinking water) and HFD for 90 days led to enlarged islet cells; reduced islet cell proliferation; and upregulated levels of Erβ, GPR30, and pro-inflammatory cytokines (TNF-α and IL-1β) in the islet cells. Based on these results, the authors suggested that BPA exposure enhances pancreatic islet cell dysfunction in postmenopausal women, but further studies are needed to support this suggestion [156].

Streptozotocin (STZ) is an antibiotic derived from *Streptomyces achromogenes*, and it damages the pancreatic β-cells and causes hypoinsulinemia and hyperglycemia. For these reasons, STZ is widely used to induce diabetes in animals [157]. Diabetic mice treated with STZ+BPA showed decreased blood glucose levels and increased insulin levels and mRNA expression of insulin transcriptional regulators *Pdx1*, *Mafa*, and *Neurod1* in pancreatic β-cells compared with those in mice treated with STZ alone [158]. Furthermore, BPA treatment increased the survival of pancreatic β-cells and insulin levels in STZ-treated mice, but serum glucose levels were not regulated. BPA-induced insulin resistance may be caused by the depletion of $Ca^{2+}$ from the cytosol and endoplasmic reticulum, inducing endoplasmic reticulum stress [159]. Additionally, BPA exposure accelerated the severity of diabetes and insulitis in non-obese diabetic mice [160–162]; decreased the number of tissue-resident macrophages in the pancreas; reduced the phagocytic activity of peritoneal macrophages [161,162]; and increased the apoptosis of α-cells, β-cells, and macrophages [161].

Although impaired insulin secretion and sensitivity are associated with an increased risk of diabetes, enhanced insulin levels and secretion rates in BPA-exposed pancreatic β-cells do not indicate increased insulin sensitivity. This is because insulin levels and secretion rates can be higher under insulin-resistance than under insulin-sensitive conditions. Furthermore, abnormally increased insulin secretion in pancreatic β-cells results in hyperinsulinemia, which is the primary driver of metabolic syndrome and related diseases such as obesity and diabetes [163,164].

### 3.4. Effects of BPA on Liver Tissues

The liver consists of hepatocytes; Kupffer cells; sinusoidal endothelial cells; stellate cells; and intrahepatic lymphocytes such as T cells, natural killer cells, and B cells. The liver is involved in the metabolism of lipids, proteins, and carbohydrates; the detoxification of foreign and toxic materials; synthesis of proteins and lipids; immune/inflammatory responses; host defense; and secretion of lipoproteins, growth factors, and cytokines [165]. BPA exposure leads to altered hepatic lipid and glucose metabolism, increased hepatic inflammatory response, and enhanced liver injury. Both free and conjugated BPA levels were found in human fetal and adult liver tissue samples [136,166–168].

For clarity, we have represented the hepatic metabolism in two sections: (1) BPA and hepatic lipid metabolism and (2) BPA and hepatic glucose metabolism. As shown in Figure 5, hepatic glucose metabolism is closely linked to hepatic lipid metabolism through several metabolic pathways.

### 3.4.1. BPA and Hepatic Lipid Metabolism

BPA treatment increases lipid content in HepG2 cells, primary mouse hepatocytes, and mouse liver tissues through various alterations in cellular responses, such as impaired autophagic degradation [169,170], hypomethylation of lipogenic genes [171], and increased pro-inflammatory phenotypes of Kupffer cells [172]. Furthermore, in vivo and in vitro studies have reported a close relationship between BPA-induced mitochondrial dysfunction and altered lipid metabolism or increased lipid accumulation [173–175].

BPA exposure during hepatic lipid metabolism increases the expression levels of genes and proteins such as SREBP1, acetyl-CoA carboxylase, fatty acid synthase, and stearoyl-CoA desaturase-1 that are involved in fatty acid biosynthesis [169,171,176–179]. BPA also increases the levels of the fatty acid transporters FAT/CD36 [179]. Moreover, BPA induces hepatic cholesterol biosynthesis mainly through the SREBP2/3-hydroxy-3-methylglutaryl-CoA reductase pathway [180–182]. These results suggest that BPA is a risk factor for accelerated progression of fatty liver disease (Figure 5).

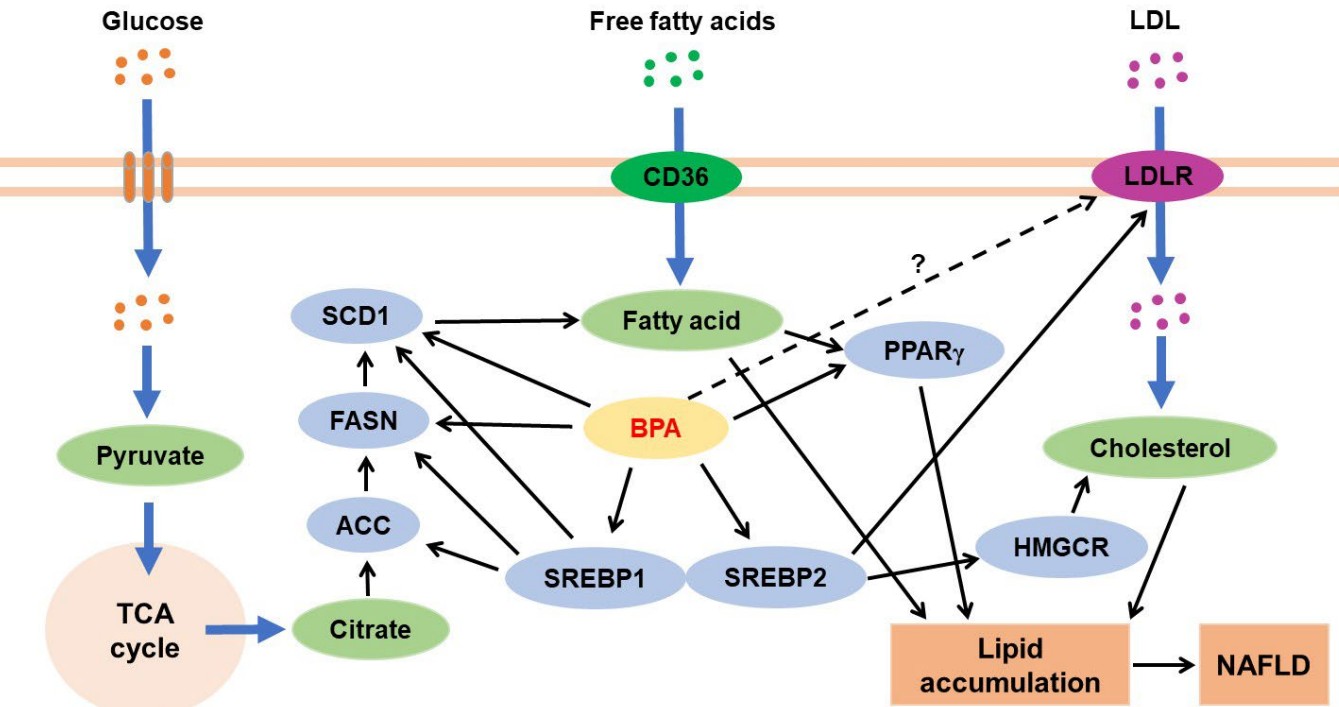

**Figure 5.** In the liver, cholesterol biosynthesis mainly occurs through the SREBP2/HMGCR pathway, and fatty acid synthesis mainly occurs through the SREBP1/ACC/FASN/SCD1 pathway. BPA induces hepatic lipid accumulation through the stimulation of these pathways. SREBP2 also stimulates the expression of LDLR [183], but whether BPA activates the SREBP2/LDLR pathway is unclear. Black arrows indicate activation cascades. ACC, acetyl-CoA carboxylase; FASN, fatty acid synthase; HMGCR, 3-hydroxy-3-methylglutaryl-CoA reductase; LDL, low-density lipoprotein; LDLR, LDL receptor; PPAR, peroxisome proliferator-activated receptor; SREBP, sterol regulatory element binding protein; SCD1, stearoyl-CoA desaturase 1; TCA, tricarboxylic acid.

Chronic consumption of an HFD increases the risk of NAFLD (typically classified as non-alcoholic fatty liver (or simple fatty liver)) characterized by hepatic steatosis; non-alcoholic steatohepatitis; and the presence of hepatocellular damage, inflammation, and fibrosis [184–186]. BPA treatment exacerbates HFD-induced hepatic dysfunction such as HFD-induced hepatic mitochondrial dysfunction [175], HFD-induced hepatic lipid accumulation [175,176,179,187], HFD-induced hepatic damage [188], and HFD-stimulated hepatic immune-metabolic dysfunction [175]. These results indicate that BPA is the leading cause of deterioration in HFD-induced liver disease.

### 3.4.2. BPA and Hepatic Glucose Metabolism

Both acute (50 µg/kg for 2 h) and chronic (50 µg/kg/day for 2 weeks) exposure to BPA impaired the activity of hepatic glucokinase, which facilitates glucose storage as glycogen [189]. After oral administration of BPA (40 µg/kg/day) in F0 pregnant dams during gestation and lactation and non-exposure of the F1 and F2 generations to BPA, the F2 generation showed glucose intolerance and insulin resistance and downregulation of the glucokinase gene in hepatic tissues [190]. In contrast, gestational exposure to BPA during embryonic day (ED) 7.5 (before the development of embryonic liver) to ED 16.5 showed no changes in hepatic glycogen content and blood glucose levels in the intraperitoneal glucose tolerance test and intraperitoneal insulin tolerance test in offspring mice [177].

Furthermore, absolute mRNA and protein levels of IR and AKT were decreased in the livers of rats treated with BPA (200 mg/kg) orally for 30 days, resulting in impaired glucose oxidation and glycogen content in the liver through defective insulin signal transduction [191]. However, in a recent study, Sprague Dawley rat dams were exposed to BPA (1 or 10 µg/mL) via drinking water from GD 6 to PND 21 and then to drinking water without BPA for 5 weeks after weaning. Following this, decreased IR and IRS protein levels but increased PI3K, AKT, and mTOR levels were observed in the liver of 8-week-old offspring [169]. These results show that further studies are required to clarify whether BPA can stimulate the PI3K/AKT/mTOR pathway in the liver because this pathway activates the SREBPs involved in lipid biosynthesis [192,193].

### 3.4.3. BPA and Liver Injury

Liver injury is caused by several factors, such as DNA damage [194], mitochondrial dysfunction caused by increased ROS levels and oxidative stress, increased inflammatory responses, reduced antioxidant defense [173,195–198], and activation of mitochondria-mediated apoptosis signaling pathways, resulting in decreased B-cell lymphoma-2 (Bcl-2) levels but increased levels of caspase-3 and -9 and Bcl-2-associated X protein (Bax) [195,196,199] (Figure 6). For example, proliferative foci and DNA damage were observed in the liver tissue of rats orally exposed to 0.5 mg/kg BPA for 90 days from PND 9 until adulthood. Enhanced ROS content plays a role in BPA-induced proliferation and DNA damage in hepatocytes [194]. Furthermore, oral perinatal exposure to BPA (50 µg/kg/day) from GD 0 to PND 21 increased hepatocyte apoptosis in the liver of rat offspring at 15 and 21 weeks of age, and the apoptotic responses were confirmed by the increased caspase-3 and caspase-9 activities and elevated levels of cytochrome c. Moreover, levels of the pro-apoptotic protein Bax increased, but that of the anti-apoptotic protein Bcl-2 decreased in offspring at 21 weeks [199]. Enhanced oxidative stress and apoptosis (due to increased caspase-3 but decreased Bcl-2 levels) were also observed in the livers of male Wistar albino rats exposed to BPA (50 mg/kg/day) by oral gavage for 8 weeks [195]. Interestingly, treatment with BPA (50 mg/kg) in wild-type mice induced the hepatic activities needed for its detoxification and elimination, whereas the same treatment in lecithin:retinol acyltransferase-deficient mice failed to induce the same activities. However, retinyl acetate supplementation restored enzyme activity and promoted BPA detoxification-mediated oxidative damage in the hepatic cells [198].

Thus, positive relationships exist between fibrosis progression and BPA-mediated mitochondrial dysfunction [175] and BPA-induced hepatic damage [195,196]. As described in our previous review, BPA is associated with regulating cancer cell growth, survival, proliferation, migration, invasion, apoptosis, and anticancer drug resistance [6]. An increased incidence of hepatocellular carcinoma (HCC) was found in 10-month-old mice exposed to BPA during the perinatal and postnatal periods [200]. However, despite NAFLD being an important cause of HCC, no reports have clarified that BPA accelerates NAFLD-triggered HCC.

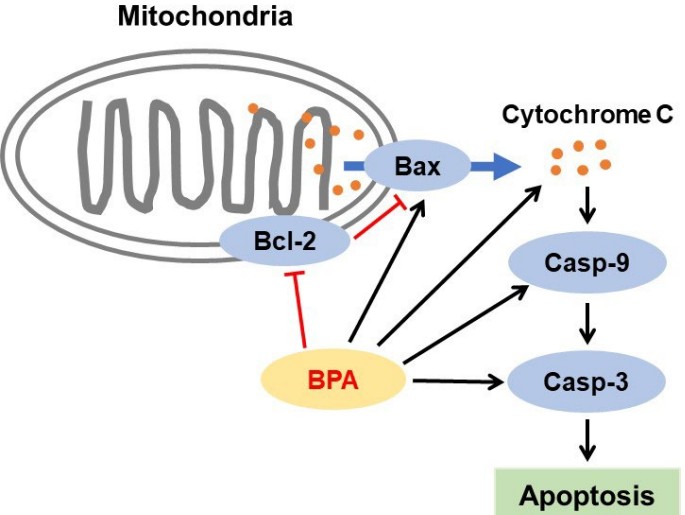

**Figure 6.** Pro-apoptotic Bax-induced release of cytochrome stimulates the caspase signaling pathway in the mitochondria, resulting in enhanced hepatic cell apoptosis. BPA exposure activates apoptosis-associated signaling pathways in hepatic cells but inhibits anti-apoptotic Bcl-2. Black arrows and red bars indicate activation and inhibition cascades, respectively. Bax, Bcl-2-associated X protein; Bcl-2, B-cell lymphoma-2; Cas, caspase.

### 3.5. Effects of BPA on the Cardiovascular System

#### 3.5.1. Effects of BPA on Heart Tissues

*Cardiac hypertrophy*: Cardiac hypertrophy is defined as the enlargement and thickening of the heart muscle and is associated with an increased risk of hypertrophic cardiomyopathy and heart failure [201,202]. Increasing evidence shows a clear association between BPA exposure and enhanced occurrence of cardiac hypertrophy. For example, when dams were exposed to BPA (25 ng/mL drinking water) from GD 11 until pup weaning on PND 21, followed by further exposure of the weaned pups to BPA (2.5 ng/mL), the 4-month-old females showed cardiac hypertrophy that was independent of increased blood pressure [203]. After daily subcutaneous injection of 0.5 mg/kg BPA in pregnant sheep during GD 30–90, increased left ventricular area and internal diameter and increased gene expression of atrial natriuretic peptide in the ventricles were observed in the 21-month-old female offspring [204]. Furthermore, combined perinatal exposure to BPA and HFD increased the interventricular septum thickness and left ventricular posterior wall thickness in matrilineal female F2 mice compared with in matrilineal female F2 mice fed a control low-fat diet [205]. In a study using human embryonic stem cell (H1, XY karyotype and H9, XX karyotype)-derived cardiomyocytes, BPA treatment (8 ng/mL) induced hypertrophic cardiomyocyte phenotypes such as elevated hypertrophic-related mRNA expression levels (e.g., NPPA and NPPB), enhanced cellular area, and reduced ATP supplementation [206]. Interestingly, BPA-exposed females showed increased sensitivity to the cardiotoxic effects of the β-adrenergic agonist isoproterenol; however, in males, BPA exposure was protective against isoproterenol-induced ischemic damage and hypertrophy compared with that in control females [207].

*Cardiomyopathy*: Long-term BPA exposure (50 µg/kg/day) of male rats for 48 weeks from the onset of delectation-induced cardiomyopathy to impaired cardiac mitochondrial function caused decreased expression and increased hypermethylation of PPARγ coactivator 1α [208]. When dams and pups were treated with BPA (2.5, 250, or 25,000 µg/kg/day) from GD 6–PND 0 and PND 1–PND 21, respectively, the females showed increased incidence and severity of cardiomyopathy, such as increased focal fibrosis and inflammatory cell infiltration at PND 21 [209].

*Myocarditis*: Myocarditis is characterized by an inflammatory condition in the heart muscle. BPA participates in the pathogenesis of myocarditis by stimulating the recruitment

of inflammatory immune cells such as macrophages, neutrophils, CD4+ T cells, and mast cells to the heart [210,211] and by increasing the levels of inflammation-related factors such as IFN-γ, IL-17A, Toll-like receptor 4, caspase-1, and IL-1β [210]. Recently, Reventun et al. suggested that BPA stimulates cardiac inflammatory responses and necroptosis by, respectively, increasing inflammatory macrophage filtration and oxidative stress and activating CaMKII and receptor-interacting protein kinase 3 necroptotic effector [212]. Additionally, BPA-mediated activation of PKA and CaMKII increased cardiac arrhythmogenesis [213]. However, progesterone administration protected against BPA-induced arrhythmogenesis through the nuclear progesterone receptor and inhibitory G protein/PI3K signaling pathway [214].

*Ischemia/reperfusion (I/R) injury*: In studies using animal models of I/R injury, BPA in combination with 17β- estradiol (E2) aggravated I/R injury through increased infarct size and severe ultrastructural disruption [215] or exacerbated ventricular arrhythmia following IR injury when compared with that observed following treatment with E2 alone [216,217].

*Cardiac fibrosis*: Cardiac fibrosis is induced by increased extracellular matrix deposition (e.g., collagen) and cardiac fibroblast activation and differentiation into myofibroblasts [218]. BPA exposure stimulates collagen deposition in the heart, which is associated with increased cardiac fibrosis [204,207,210,211,219–221]. Moreover, BPA treatment increased collagen production in cardiac fibroblasts by activating ERK1/2 [222] but reduced cardiac remodeling after MI by increasing macrophage-induced inflammation and reducing myofibroblast repair function [203]. Additionally, combined treatment of BPA + BPS + BPAF (1 or 10 ng/mL of each) increased the collagen levels during embryonic stem cell differentiation into cardiomyocytes via the p38 pathway, but not when treated with BPA alone; BPS and BPAF are BPA derivatives [223].

Taken together, ERK1/2 and CaMKII pathways play critical roles in BPA-mediated cardiac fibrosis and dysfunction (Figure 7).

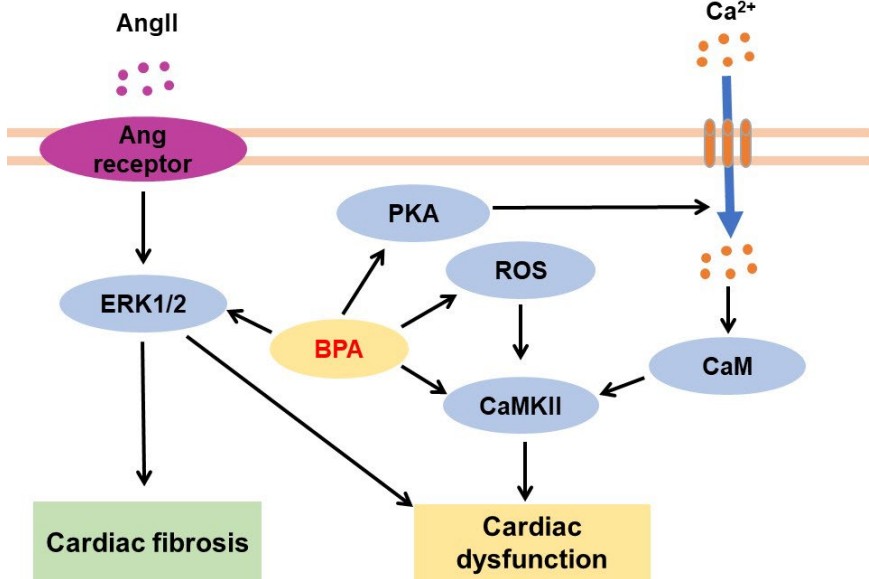

**Figure 7.** ERK1/2 and CaMKII pathways are associated with BPA-mediated cardiac fibrosis and dysfunction. Black arrows indicate activation cascades. Ang, angiotensin; CaM, Ca$^{2+}$/calmodulin; CaMKII, Ca$^{2+}$/calmodulin-dependent protein kinase II; ERK1/2, extracellular signal-regulated protein kinase 1/2; PKA, protein kinase A; ROS, reactive oxygen species.

### 3.5.2. Effects of BPA on Blood Vessels

*Angiotensin and blood pressure*: BPA exposure increases the levels of angiotensin (Ang) [224,225], Ang-converting enzyme [226], and angiotensinogen (a precursor of Ang) [220], which play a critical role in cardiovascular physiology [227]. When post-weaned male mice were fed a low-protein diet for 8 weeks and then exposed to BPA (50 μg/kg/day)

for the last 9 days, higher angiotensinogen mRNA expression was found in mice treated with a low-protein diet + BPA than in those fed a low-protein diet or BPA alone [220]. BPA treatment increased arterial AngII levels [225] and vascular smooth muscle cell proliferation through AngII-induced ERK1/2 activation [224] and stimulated the activation of the Ang-converting enzyme under the hypertensive milieu [226]. Furthermore, BPA induces high blood pressure (hypertension) by reducing eNOS levels [205], increasing AngII/CaMKII-$\alpha$ uncoupling of eNOS [225], or enhancing vascular ROS/NO imbalance [228].

*Atherosclerosis*: BPA exposure leads to increased atherosclerotic lesions in the aortas of ApoE-deficient mice via the induction of non-HDL cholesterol levels [229] or CD36 expression and lipid accumulation in macrophages [230]; in the aortas of LDL receptor-deficient mice via a reduction in sirtuin 1-mediated DNA repair [231]; and in the aortas of Watanabe heritable hyperlipidemic rabbits via the induction in inflammatory responses and metabolic disorders [232,233].

Atherosclerosis is a risk factor for vascular calcification, which mainly occurs in vascular smooth muscle cells through several signaling pathways, such as PKA and p38 MAPK [234–236]. The mechanisms underlying vascular calcification and bone calcification (or mineralization) are similar [237,238], and BPA exposure results in increased bone calcification [239,240]. However, despite these reports, no direct evidence has been reported for BPA exposure-induced vascular calcification.

### 3.6. Effects of BPA on Kidney Tissues

Podocytopathy is a kidney disease characterized by podocyte injury-mediated proteinuria or nephrotic syndrome [241]. BPA exposure increases the risk of podocytopathy [242] or podocyte injury in immortalized human podocytes [243]. In human renal proximal tubular epithelial cells (HK-2 and ATCC CRL-2190), acute BPA exposure (24 h) increases mitochondrial dysfunction, oxidative stress, and apoptotic death in a concentration-dependent manner [244]. Furthermore, rhesus monkey embryo renal epithelial Marc-145 cells exposed to BPA exhibited increased oxidative stress, apoptosis, and DNA damage [245].

BPA treatment promotes kidney injury in rats and mice with or without diabetes [243,246] through increased oxidative stress [247], apoptosis [246–248], inflammatory responses [243,249], and impaired mitochondrial function in the kidney [247]. In rats, oral administration of BPA (50 mg/kg) led to worsened renal injury induced by ischemia-reperfusion, including increased renal dysfunction and histopathological abnormalities, oxidative stress, apoptosis, mitochondrial functional impairment, mitochondrial dynamics, and mitophagy disproportion. The exacerbated effects of BPA were associated with the stimulation of the AMPK/PGC-1$\alpha$/SIRT3 axis [247]. In MRL/lpr mice that were orally exposed to BPA, exacerbating lupus nephritis was associated with increased inflammatory responses, abnormal autophagy, and decreased antioxidant capacity [250]. Moreover, intraperitoneal injection of 120 mg/kg/day of BPA (5 days a week) in subtotal nephrectomy mice for 5 weeks induced renal damage and fibrosis and blocked autophagosome maturation in the kidney [249]. Pregnant mice treated with BPA (10 or 100 µg/kg/day) during GD 9–16 (early nephrogenesis) showed glomerular abnormalities and reduced glomerular formation in the 30-day-old offspring, indicating that BPA exposure during embryonic development can alter nephrogenesis [251]. A recent study has suggested that BPA-induced ferroptosis in renal tubular epithelial cells (TCMK-1) depends on the activation of ferritinophagy through the AMPK/mTOR/ULK1 axis [252].

### 3.7. Effects of BPA on Intestinal and Gut Microbiota

The gut microbiota performs key metabolic activities and communicates with organ systems through different means, such as the gut–liver axis [253,254] or gut–brain axis [255]. The disruption of normal gut flora by BPA can impact metabolic pathways, intestinal permeability, and barrier function. BPA exposure (5 µg/kg/day) during GD 19–21 affected the gut microbial composition, which potentially plays a key role in the interactions with the liver metabolic pathways and liver genes involved in oxidative phosphorylation, PPAR

signaling, and fatty acid metabolism [253]. After BPA treatment, reduced levels of short-chain fatty acid (SCFA) producers such as *Oscillospira* and *Ruminococcaceae* led to decreased fecal SCFA levels, increased systemic lipopolysaccharide levels, and altered amino acid metabolism [256]. BPA also increases the number of metabolites involved in carbohydrate metabolism and synthesis in females and alters lysine degradation and phenylalanine and tyrosine metabolism in males [257].

Furthermore, BPA disrupts gut barrier function by reducing the number of intestinal goblet cells and mucin 2 gene expression [258], inducing apoptosis of intestinal epithelial cells and reducing their proliferation [259], and inducing dysbiosis of the gut microbiota [254,260]. BPA-fed mice showed an increased abundance of *Proteobacteria* (a marker of dysbacteria) and a decreased abundance of *Akkermansia* (a gut microbe associated with increased gut barrier function and reduced inflammation) [254]. BPA-mediated disruption of the gut barrier function increases intestinal permeability [254,258–260].

In the natural environment, BPA is biodegraded by various microorganisms, including bacteria and fungi [4,5,261]. Similarly, several numbers of the gut microbiota such as *Microbacterium*, *Alcaligenes*, Sporobiota (spore-forming bacteria), *Bacillus* spp., and *Bacillus* sp. AM1 are capable of biodegrading BPA [262–264]. Interestingly, microorganisms from the gut microbiota of children with obesity showed higher BPA biodegradation potential than those from children with normal weight [263]. However, further studies are required to clarify whether BPA-biodegradable gut microbiota affect obesity.

*3.8. BPA and Exercises*

A recent study suggested that BPA can be removed more efficiently in young males than in females after physical exercise [265]. Furthermore, the female offspring of BPA-exposed female mice that were fed diets supplemented with 50 mg/kg BPA 2 weeks before breeding until lactation showed a higher average respiratory quotient during the dark and light cycles. This indicates carbohydrate rather than fat metabolism and a significantly lower activation during the dark cycle compared to control female offspring, while there were no differences in serum glucose, insulin, adiponectin, and leptin concentrations. However, BPA-exposed males failed to exhibit similar effects to those in BPA-exposed females [266].

## 4. BPA Exposure and Alternation of Lipid Compositions

Among 744 lipid species in the serum lipidome, the serum TG species showed the strongest association with urinary BPA levels in healthy young adults [267]. Perinatal BPA exposure altered the composition of milk lipids. Saturated FA and C18:1n-9 monosaturated FA levels were higher in the group exposed to BPA (250 µg/L in drinking water) from GD 9 until weaning than in the control group [268]. As mentioned earlier, BPA exposure (0.5 µg/kg) decreased the levels of the essential polyunsaturated FA linoleic acid (18:2n-6) in iWAT and liver TG in male offspring. However, the SCD-1 levels or FA composition were not altered in 5-week-old female offspring exposed to maternal BPA (0.5 and 50 µg/kg) [113]. A lipid profiling analysis revealed that total FA, acylcarnitine, and monoacylglycerol levels increased on PND 21 in both female and male rats exposed to BPA (5000 µg/kg/day). Enhanced total cholesterol ester levels and reduced triacylglycerol and monogalactosyl diacylglycerol levels were found in PND 21 females [269]. High maternal BPA levels during pregnancy were associated with an increase in neonatal mono-unsaturated alkyl-lysophosphatidylcholine (LPC) levels in umbilical venous cord blood samples [270]. In contrast, treatment with BPS (a BPA analog) increased LPC, phosphatidylethanolamine (PE), lysophosphatidylethanolamine (LPE), and lysophosphatidylserine (LPS) levels but reduced those of phosphatidylcholine and phosphatidylserine (PS) in the mouse liver [271]. In the spleen, it increased PS, LPC, LPE, sphingosine, neutral glycosphingolipids, cholesteryl ester, diacylglycerol, phosphatidylglycerol, and phosphatidylinositol levels but decreased PE, ceramide, and sphingomyelin levels [272].

These alterations in lipid composition may aggravate cardiovascular or cardiometabolic diseases because LPS and LPC are positively associated with inflammatory response, vascular cell calcification, and apoptosis [273–275]. However, very little data show a relationship between the alteration of BPA-induced lipid composition and cardiovascular or cardiometabolic diseases.

## 5. Factors Influencing BPA-Mediated Disorders in Cardiovascular or Cardiometabolic Systems

Several in vivo and in vitro studies have suggested that various factors influence the metabolic or toxic effects of BPA on various cardiovascular and cardiometabolic systems. In this section, some factors that should be carefully considered for a better understanding of BPA-mediated disorders are described.

*Sex difference*: BPA exposure results show different disorders between males and females in a sex-specific manner. For example, urinary BPA levels are positively associated with fasting hyperglycemia and β-cell dysfunction in women but not in men [10]. BPA treatment resulted in increased glomerular damage and oxidative stress in the kidneys, which was more pronounced in male mice than in females [246]. Interestingly, BPA exposure tends to induce disorders with more severity in the adipose tissues of males than in females (see Section 3.2). Therefore, sex differences should be carefully considered when studying the relationships between BPA and various cardiovascular or cardiometabolic diseases or the potential mechanisms underlying BPA-mediated disorders in these diseases.

*Species*: BPA induces different responses among different species. For instance, the BPA-induced reduction in $K_{ATP}$ channel activity and insulinotropic effect are stronger in humans than in mouse β-cells [151]. Moreover, the functional roles of UDP-glucuronosyltransferase enzymes expressed in the liver and intestines differ extensively among humans, monkeys, dogs, rats, and mice [276]. The hepatic capacity for BPA glucuronidation is predicted to be in the order of humans > rats > mice [277]. Murine animal models, such as mice and rats, are useful for understanding BPA-mediated dysfunction in various cardiovascular or cardiometabolic systems. However, recognizing that BPA induces different responses in mice and humans is essential.

*BPA concentrations*: High BPA concentrations do not always lead to serious disorders in cardiovascular or cardiometabolic systems. For example, significantly increased SCD-16 and SCD-18 indices in iWAT TG and plasma cholesterol esters were found in 5-week-old male offspring treated with 0.5 μg/kg rather than with 50 μg/kg BPA [113]. Furthermore, compared with that in negative controls, male newborns exposed to 5 or 500 μg/kg/day of BPA exhibited a greater increase in gonadal fat pad weight, gonadal fat cells, and adipocyte volume than newborns exposed to 5000 μg/kg/day of BPA [114]. In addition, when rats were exposed to BPA (0.5 or 50 μg/kg/day) via drinking water from GD 3.5 until PND 22, insulin secretion was enhanced in islets isolated from 5- and 52-week-old female and male offspring and from dams treated with 0.5 μg/kg/day of BPA but was reduced after treatment with 50 μg/kg/day of BPA [144]. Furthermore, increased blood pressure and plasma glucose levels were higher in female children with high prenatal BPA levels than in those with low prenatal BPA levels [278]. These results indicate that in vitro or in vivo evaluation using various BPA concentrations may provide a deeper understanding of BPA-mediated disorders in various cardiovascular and cardiometabolic systems.

*BPA exposure or investigation timing*: Growing evidence shows that BPA exposure or investigation timing is an important factor to consider when studying the effects of BPA on various cardiovascular or cardiometabolic systems. For instance, after BPA exposure in pregnant mice, their offspring showed an increase in pancreatic β-cell mass and proliferation but a decrease in β-cell apoptosis at PND 0, PND 21, and PND 30; however, equal or decreased β-cell mass was observed at PND 120 [146]. BPA exposure during prenatal periods rather than postnatal periods seems to induce more enhanced disorders in the cardiovascular or cardiometabolic systems. When BPA (100 μg/kg/day) was administered at preimplantation (days 1–6 of pregnancy; P 1–P 6), fetal (P 6–PND 0), neonatal (PND

0–PND 21), or fetal and neonatal (P 6–PND 21) periods, impaired glucose homeostasis was found in P 6–PND 0 mice between the ages of 3 and 6 months, and this continued up to 8 months in males. However, in the PND 0–PND 21 and P 6–PND 21 BPA-treated groups, only the 3-month-old male offspring developed glucose intolerance [279].

*Combined exposure to BPA and other agents*: Several studies have reported that combined exposure to BPA and other harmful agents induces more severe disorders or toxicity in the cardiovascular or cardiometabolic systems than exposure to BPA alone due to a synergistic effect. Combined exposure to BPA (100 µg/L) and perfluorooctane sulfonate (PFOS; 2000 µg/L) for 19 days during pregnancy induced morphological changes in the fetal rat heart compared with PFOS or BPA exposure alone. The combination of PFOS (100 ng/mL) and BPA (10 ng/mL) for 14 days during the cardiac differentiation period increased cardiomyocyte size and collagen content compared with PFOS or BPA exposure alone [280]. Moreover, after exposure of male Wistar albino rats to BPA (5 mg/kg), lead (Pb) (100 ppm), endosulfan (0.61 mg/kg), BPA and Pb, BPA and endosulfan, Pb and endosulfan, and all three (BPA + Pb + endosulfan) for 65 days, the BPA + Pb + endosulfan combination group exhibited more severe histopathological changes in the liver and kidney tissues; elevated malondialdehyde in the liver; decreased superoxide dismutase activity in the kidney tissue; and increased serum ALT, BUN, and creatinine levels compared to the control group [281]. In addition, co-exposure to BPA (10 µg/kg) and arsenic (10 ppb) activated the genes involved in glycogenesis, glucogenesis, and fatty acid oxidation in the liver [282].

In addition, in the case of combined exposure to other endocrine-disrupting agents, low BPA levels are sufficient to increase the effects of the disruptors of cardiovascular or cardiometabolic systems. For example, combined treatment with BPA + BPS + BPAF (each 1 ng/mL) increased collagen levels during embryonic stem cell differentiation into cardiomyocytes, but treatment with 10 ng/mL of BPA alone did not [223]. Furthermore, endocrine disruptors with a similar structure to BPA can also induce disorders or dysfunctions in the tissues associated with various cardiovascular or cardiometabolic diseases. As an example, the synthetic estrogen diethylstilbestrol (DES), which exhibits a high binding affinity for the ER, increases the risk of cardiovascular or cardiometabolic diseases [203,283–285]. However, further studies are needed to clarify whether combined exposure to BPA and DES causes more severe cardiovascular toxicities or disorders than exposure to BPA alone.

## 6. Effects of BPA on the Efficacy of Therapeutic Drugs for Cardiovascular or Cardiometabolic Diseases

Many studies have suggested that BPA decreases the efficacy of some common anticancer drugs used in cancer treatment. The mechanisms underlying BPA exposure in terms of reducing the efficacy of anticancer drugs have been broadly investigated [286,287]. These studies also suggest that BPA reduces the efficacy of chemotherapeutic drugs used to treat or protect against cardiovascular or cardiometabolic diseases. For example, estrogen therapy with E2 can protect against fatty liver, insulin resistance, and diabetes in premenopausal women [288]. However, a recent study has suggested that BPA disrupts E2-mediated hepatic protection in I/R injured liver by upregulating the Ang II/AT1R signaling pathway [289], indicating that BPA decreases E2 efficacy against liver damage. E2 can also increase cardioprotection against I/R injury by improving mitochondrial function and reducing ROS levels, mainly through ERα and GPR30 [290]. However, the combined treatment with E2 and BPA led to aggravated I/R injury and enhanced ultrastructural disruption [215] or exacerbated ventricular arrhythmia following IR injury compared with that observed after treatment with E2 alone [216,217].

Furthermore, as mentioned in Section 3.5.2, BPA increases arterial AngII levels and stimulates Ang-converting enzymes and AngII-mediated signal transduction pathways. Ang-converting enzyme inhibitors and AngII receptor antagonists are widely used to prevent or manage cardiovascular diseases [291,292]. However, very few studies have

examined whether the efficacy of these drugs is influenced by BPA-induced activation of the Ang-converting enzyme and AngII-related signals.

Furthermore, among persons with type I or II diabetes, those with insulin resistance may require higher doses of insulin to control blood glucose levels than those with higher insulin sensitivity. As mentioned earlier (Section 3.3), BPA exposure increases the risk of insulin resistance. However, very little information is available on whether BPA-induced insulin resistance is associated with an increase in insulin dosage or frequency of injections.

Taken together, further research is needed to understand the mechanisms underlying the relationship between BPA exposure and reduced efficacy of chemotherapeutic drugs used for the treatment of cardiovascular or cardiometabolic diseases.

## 7. Summary and Overall Conclusions

The main cause for high BPA levels in human serum and urine is the uptake of BPA-contaminated foods [1,2]. Growing evidence suggests that serum or urinary BPA levels are positively associated with an increased risk of cardiovascular or cardiometabolic diseases such as diabetes; obesity; hypertension; and heart and kidney diseases such as AKD, CKD, myocardial MI, stroke, and CAD. Therefore, refraining from the consumption of BPA-contaminated foods can reduce the risk factors for various cardiovascular or cardiometabolic diseases.

Furthermore, most available data pertain to healthy adults, but very little data are available for pregnant and lactating women. However, many animal experiments have suggested that compared with BPA exposure during childhood or adulthood, exposure during pregnancy or lactation is much more likely to induce the development of cardiovascular or cardiometabolic diseases. Additionally, the identification of cardiovascular or cardiometabolic disease-related genes, lipids, and proteins in the urine and blood of patients with elevated BPA concentrations would help to gain a deeper understanding of the mechanism underlying the response to BPA.

BPA induces disorders or dysfunction in body tissues associated with cardiovascular or cardiometabolic diseases through a variety of cell signaling pathways. For example, BPA decreases glucose uptake in skeletal muscles by inhibiting the AKT/AS160/GLUT4 pathway. In adipocytes, BPA-mediated inflammatory responses are stimulated through the JAK/STAT3, IKK/NF-κB, and JNK/AP-1 axes. Furthermore, biosynthesis of cholesterol and fatty acids in the liver is controlled through the SREBP2/HMGCR and SREBP1/ACC/FASN/SCD1 pathway, respectively. ERK1/2 and CaMKII pathways play critical roles in BPA-mediated cardiac fibrosis and dysfunction. However, these are just a few of the signaling pathways involved in BPA-mediated disorders or dysfunction of cardiovascular cells or tissues.

Finally, more research needs to be undertaken to resolve several remaining questions, such as the link between BPA-mediated cardiovascular or cardiometabolic diseases and the effect of BPA on the efficacy of chemotherapeutic drugs, BPA-induced alteration of lipid compositions, and BPA-related autoimmune mechanisms.

**Author Contributions:** Conceptualization, J.-H.K.; methodology, J.-H.K.; formal analysis, J.-H.K. and R.T.; investigation, J.-H.K., D.A. and R.T.; writing—original draft preparation, J.-H.K. and R.T.; writing—review and editing, J.-H.K., D.A. and R.T.; supervision, J.-H.K.; funding acquisition, J.-H.K. All authors have read and agreed to the published version of the manuscript.

**Funding:** This study was supported by the National Cerebral and Cardiovascular Center Research Institute KIBANKEIHI (fiscal year 2023).

**Institutional Review Board Statement:** Not applicable.

**Informed Consent Statement:** Not applicable.

**Data Availability Statement:** Not applicable.

**Conflicts of Interest:** The authors declare no conflict of interest.

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
