# Peer review of "Bisphenol A (BPA) and Cardiovascular or Cardiometabolic Diseases"

_jox, doi:10.3390/jox13040049_

Round 1
Reviewer 1 Report
Comments and Suggestions for Authors
The review by Kang et al. is a detailed, and fairly well-supported compilation of research results from studies of BPA exposure and its impact on cardiovascular/cardiometabolic diseases. The review is well organized and clearly written. The authors provided a succinct summary of the often contradictory findings of studies between males and females, different species, exposure levels, and influence of time of exposure.
The authors failed to describe the method used to collect research studies for the review, however. How did the authors go about finding articles? How were publications selected to be included? For example, was a specific date range considered? The citations seem to be biased towards publications from 2000-2018 with few citations from the past 5 years.
The authors provide a great deal of evidence of negative health impacts related to BPA exposure, but leave out context to help the reader assess the extent of the problem. The review would be improved with information about how governments, corporations, and consumers are responding to evidence of health effects of BPA exposure.
Author Response
We would like to thank the reviewer for all useful suggestions and comments on our manuscript (Manuscript ID: jox-2695948). We have carefully considered the reviewer’s suggestions and comments and have revised our manuscript.
Our response to the reviewer’s suggestions and comments is as follows:
Reviewer 1
Comment 1) The review by Kang et al. is a detailed, and fairly well-supported compilation of research results from studies of BPA exposure and its impact on cardiovascular/cardiometabolic diseases. The review is well organized and clearly written. The authors provided a succinct summary of the often contradictory findings of studies between males and females, different species, exposure levels, and influence of time of exposure.
The authors failed to describe the method used to collect research studies for the review, however. How did the authors go about finding articles? How were publications selected to be included? For example, was a specific date range considered? The citations seem to be biased towards publications from 2000-2018 with few citations from the past 5 years.
Response) Thank you very much for your helpful comments. The literatures related to bisphenol A (BPA) and cardiovascular or cardiometabolic diseases were searched using PubMed and Google Scholar databases. The following keywords were used for the literature search: “bisphenol A” AND “kidney”, “bisphenol A” AND “liver”, “bisphenol A” AND “heart”, “bisphenol A” AND “pancreatic tissue”, “bisphenol A” AND “skeletal muscle”, “bisphenol A” AND “urine”, “bisphenol A” AND “blood”, “bisphenol A” AND “plasma” “bisphenol A” AND “diabetes”, “bisphenol A” AND “obesity”, “bisphenol A” AND “hypertension”, “bisphenol A” AND “adipogenesis”, “bisphenol A” AND “adipocyte”, “bisphenol A” AND “adipose tissue”, “bisphenol A” AND “inflammation”, “bisphenol A” AND “cytokine”, “bisphenol A” AND “insulin resistance”, “bisphenol A” AND “lipid metabolism”, “bisphenol A” AND “glucose metabolism”, “bisphenol A” AND “cardiac hypertrophy”, “bisphenol A” AND “cardiomyopathy”, “bisphenol A” AND “myocarditis”, “bisphenol A” AND “cardiac fibrosis”, “bisphenol A” AND “angiotensin”, “bisphenol A” AND “blood vessel”, “bisphenol A” AND “gut microbiota”, “bisphenol A” AND “exercise”, and “bisphenol A” AND “lipid composition”. Finally, 293 articles from 2000 until March 2023 were adopted in the present manuscript.
Furthermore, among 293 references, the past 5 years papers (2018-2022) cited in our manuscript are 151 (51.1%), indicating that this is by no means small.
However, we would not like to add the section of Materials and Methods related to the literature search. We really hope the readers to fix on the content of our manuscript and to find new research ideas and topics. We also wish the readers to reduce time and effort to find articles based on our method for the literature search.
Comment 2) The authors provide a great deal of evidence of negative health impacts related to BPA exposure, but leave out context to help the reader assess the extent of the problem. The review would be improved with information about how governments, corporations, and consumers are responding to evidence of health effects of BPA exposure.
Response) We really appreciate your helpful comments. As mentioned in the Introduction, the major route of exposure is through BPA-contaminated foods. Therefore, refraining from consumption of BPA-contaminated foods can reduce the risk factors for various cardiovascular or cardiometabolic diseases. Based on your wonderful comments, we have added the following sentence to the text:
Therefore, refraining from consumption of BPA-contaminated foods can reduce the risk factors for various cardiovascular or cardiometabolic diseases.
I have marked all corrections and alterations in red.
I believe the manuscript has been improved satisfactorily and meets the publication in Journal of Xenobiotics.
Reviewer 2 Report
Comments and Suggestions for Authors
In this review concerning the effects of BPA on the onset of cardiovascular and cardiometabolic disorders, the authors make an excellent contribution to scientific knowledge. However, a few details and suggestions should be taken into account:
Abstract line 16, Introduction line 43 and conclusion line 1043: reverse obesity and diabetes to respect the order chosen by the authors in successive chapters.
In the introduction the authors could have referred to the synthetic estrogen diethylstilbestrol (DES) of an equivalent chemical formula and to work concerning the same cardiovascular disorders, for example by citing:
-Troisi et al., A prospective cohort study of prenatal DES exposure and cardiovascular disease risk. 2018, J.Clin. Endocrinol. Metab., 103 (1) 206-212.
-See also: Haddad R. et al., 2013, Can J Physiol. 91 (9), 741-749
-Patel et al., 2015, Toxicol Rep 2, 1310-1319
-Yi-Feng Li et al., 2019, Environ Int 124, 511-520
Introduction line 112: a question: Why these different results? problem of methodology?
2.
2.3. BPA level and hypertension: line 181,182, ref 57: do the authors of this article have an explanation?
Is there any work regarding BPA detoxification and Cytochromes P450?
The sentence "For clarity.....glucose metabolism" lines 714-715, must be replaced just before section 3.4.1. and not after.
About figures: They are excellent and attempt to shed light on the complexity of action of this toxicant at certain key points. Since these are the authors who made them and these are not quotes, it would be good to point this out at the end of the introduction.
It would also be interesting to summarize all the deleterious actions of BPA at the cardiovascular and cardiometabolic levels in a summary table.
This article deserves to be published in JOX, with the congratulations of the reviewer.
Comments on the Quality of English Language
No comment
Author Response
We would like to thank the reviewer for all useful suggestions and comments on our manuscript (Manuscript ID: jox-2695948). We have carefully considered the reviewer’s suggestions and comments and have revised our manuscript.
Our response to the reviewer’s suggestions and comments is as follows:
Reviewer 2
In this review concerning the effects of BPA on the onset of cardiovascular and cardiometabolic disorders, the authors make an excellent contribution to scientific knowledge. However, a few details and suggestions should be taken into account:
Comment 1) Abstract line 16, Introduction line 43 and conclusion line 1043: reverse obesity and diabetes to respect the order chosen by the authors in successive chapters.
Response) Thank you very much for your kind comment. According to your comment, we have changed the order as follows: diabetes, obesity.
Comment 2) In the introduction the authors could have referred to the synthetic estrogen diethylstilbestrol (DES) of an equivalent chemical formula and to work concerning the same cardiovascular disorders, for example by citing:
Troisi et al., A prospective cohort study of prenatal DES exposure and cardiovascular disease risk. 2018, J.Clin. Endocrinol. Metab., 103 (1) 206-212.
-See also: Haddad R. et al., 2013, Can J Physiol. 91 (9), 741-749
-Patel et al., 2015, Toxicol Rep 2, 1310-1319
-Yi-Feng Li et al., 2019, Environ Int 124, 511-520
Response) Based on your useful comments, the following sentences have been added to the section of Combined exposure to BPA and other agents:
Furthermore, endocrine disruptors with a similar structure to BPA can also induce disorders or dysfunctions in the tissues associated with various cardiovascular or cardiometabolic diseases. As an example, the synthetic estrogen diethylstilbestrol (DES), which exhibits a high-binding affinity for the ER, increases the risk of cardiovascular or cardiometabolic diseases [282-285]. However, further studies are needed to clarify whether combined exposure to BPA and DES causes more severe cardiovascular toxicities or disorders than exposure to BPA alone.
- Troisi, R.; Titus, L.; Hatch, E.E.; Palmer, J.R.; Huo, D.; Strohsnitter, W.C.; Adam, E.; Ricker, W.; Hyer, M.; Hoover, R.N. A prospective cohort study of prenatal DES exposure and cardiovascular disease risk. J. Clin. Endocrinol. Metab. 2018, 103, 206–212.
- Haddad, R.; Kasneci, A.; Sebag, I.A.; Chalifour, L.E. Cardiac structure/function, protein expression, and DNA methylation are changed in adult female mice exposed to diethylstilbestrol in utero. Can J Physiol Pharmacol. 2013, 91, 741–749.
- Patel, B.B.; Raad, M.; Sebag, I.A.; Chalifour, L.E. Sex-specific cardiovascular responses to control or high fat diet feeding in C57bl/6 mice chronically exposed to bisphenol A. Toxicol Rep. 2015, 2, 1310–1318.
- Li, Y.F.; Canário, A.V.M.; Power, D.M.; Campinho, M.A. Ioxynil and diethylstilbestrol disrupt vascular and heart development in zebrafish. Environ Int. 2019, 124, 511–520.
Comment 3) Introduction line 112: a question: Why these different results? problem of methodology?
Response) We have checked the content of all references cited in our manuscript once more. However, we honestly don’t know the reason why different results have been obtained because of various factors such as different age, gender, country, sampling time and method, analytical method, etc.
Comment 4) 2.3. BPA level and hypertension: line 181,182, ref 57: do the authors of this article have an explanation?
Response) The authors did not give any explanation on no interaction between serum BPA and estradiol levels. However, we have added three words to clarify the fourth quartile of women as follows:
Additionally, based on the serum BPA levels, the fourth quartile of women (mainly postmenopausal women) exhibited a higher risk of hypertension than those in the lowest quartile. However, no interaction was observed between serum BPA and estradiol level [57].
Comment 5) Is there any work regarding BPA detoxification and Cytochromes P450?
Response) We deeply appreciate your comment. It is a fact that several studies regarding BPA detoxification and Cytochromes P450 (mainly elucidation of detoxification mechanisms or pathways) have been reported. But as far as we know, there are no reports on the relationship between BPA detoxification by cytochromes P450 and disorder or toxicity in the cardiovascular or cardiometabolic systems.
Comment 6) The sentence "For clarity.....glucose metabolism" lines 714-715, must be replaced just before section 3.4.1. and not after.
Response) As suggested, we have removed the sentence before the section of 3.4.1. BPA and hepatic lipid metabolism (please see the revised manuscript).
Comment 7) About figures: They are excellent and attempt to shed light on the complexity of action of this toxicant at certain key points. Since these are the authors who made them and these are not quotes, it would be good to point this out at the end of the introduction.
Response) Thank you very much for your kind valuable comments. The following sentence has been added to the last line of Introduction according to your comments:
Furthermore, BPA-mediated disorders or dysfunctions and their related signaling pathways are summarized in Figure within each section.
Comment 8) It would also be interesting to summarize all the deleterious actions of BPA at the cardiovascular and cardiometabolic levels in a summary table.
Response) As our respond to Comment 7, in fact, BPA-mediated disorders or toxicities and their related signaling pathways in various cardiovascular or cardiometabolic diseases have been summarized in Figure within each section. We are very sorry, but we would like you to understand that we have not added a novel summary Table to avoid duplication.
I have marked all corrections and alterations in red.
I believe the manuscript has been improved satisfactorily and meets the publication in Journal of Xenobiotics.